# G-TRACER: Expected Sharpness Optimization

## Abstract

We propose a new regularization scheme for the optimization of deep learning architectures, G-TRACER ("Geometric TRACE Ratio"), which promotes generalization by seeking minima with low mean curvature, and which has a sound theoretical basis as an approximation to a natural gradient-descent based optimization of a generalized variational objective. By augmenting the loss function with a G-TRACER penalty, which can be interpreted as the metric trace of the Hessian (the Laplace-Beltrami operator) with respect to the Fisher information metric, curvature-regularized optimizers (e.g. SGD-TRACER and Adam-TRACER) are simple to implement as modifications to existing optimizers and do not require extensive tuning. We show that the method can be interpreted as penalizing, in the neighborhood of a minimum, the difference between the mean value of the loss and the value at the minimum, in a way that adjusts for the natural geometry of the parameter space induced by the KL divergence. We show that the method converges to a neighborhood (depending on the regularization strength) of a local minimum of the unregularized objective, and demonstrate promising performance on a number of benchmark computer vision and NLP datasets, with a particular focus on challenging problems characterized by a low signal-to-noise ratio, or an absence of natural data augmentations and other regularization schemes.

## 1 Introduction

Contemporary neural network architectures (e.g. Llama 2: 70B parameters, GPT-4: 1.7T parameters) are typically overparameterized, with more parameters than constraints (Liu et al., 2022). The fact that interpolating solutions with no explicit regularization can generalize well to unseen data (Zhang et al., 2016) (Belkin et al., 2019) is surprising from a classical statistical learning perspective, and there is an emerging consensus that this phenomenon is due to implicit regularization in which, in very high dimensional settings, among all interpolating solutions, well-behaved minimum-norm solutions are preferred[1] (Curth et al., 2023). Whether implicit regularization alone suffices is problem-dependent and is influenced, among many other factors, by the signal-to-noise ratio (Hastie et al., 2022). In practical settings, non-zero weight decay is typically applied, and explicit regularization is key to obtaining SOTA performance.[2]

Deep neural networks possess discrete and continuous symmetries (transformations that leave the underlying function invariant (Kristiadi et al., 2023)) as well as reparameterization invariance with respect to many common coordinate changes (e.g. BatchNorm (Ioffe & Szegedy, 2015), WeightNorm (Salimans & Kingma, 2016)). There is a large literature relating these characteristics of deep neural networks to the geometry of the loss surface[3] (Liu et al., 2022). Li et al. (2018) show, under mild assumptions, that sufficiently wide networks have no set-wise local minima that are not global minima. Moreover, solutions to overparameterized neural networks typically form a high-dimensional manifold (Cooper, 2018) and are characterized by degenerate Hessians (Sagun et al., 2017), where the bulk of the eigenvalue spectrum is clustered around 0.

The connection between the geometry of the loss surface and generalization has long been the subject of interest and speculation, dating back to the MDL-based arguments of Hinton & van Camp (1993) and

---

[1]Indeed this effect can be observed even in certain linear models in the overparameterized regime $p > n$ (Hastie et al., 2022)

[2]Weight decay is equivalent ridge regularization, and it can be shown that minimum l2-norm regression is a limiting case of ridge regression as the ridge penalty goes to 0.

[3]Viewing the loss, for example, as a hypersurface: $\{(\boldsymbol{w}, L(\boldsymbol{w})), \boldsymbol{w} \in \mathbb{R}^{p+1} : \boldsymbol{w} \in \Theta\}$

Hochreiter & Schmidhuber (1997). In particular, the connection between sharpness and generalization is an intuitively appealing one, in that the sharp local minima of the highly nonlinear, non-convex optimization problems associated with modern large-scale deep learning architectures are more likely to be brittle and sensitive to perturbations in the parameters and training data, and thus lead to worse performance on unseen data. The recent success of the SAM algorithm, which measures sharpness as $\max_{\|\boldsymbol{\Delta w}\|_2 \leq \epsilon} L(\boldsymbol{w} + \boldsymbol{\Delta w}) - L(\boldsymbol{w})$ (Foret et al., 2020) has reignited interest in geometrically motivated regularization schemes. We propose a novel regularization scheme, which implicitly measures Sharpness as $\mathrm{Tr}(\boldsymbol{G}^{-1}\boldsymbol{H})$, where $\boldsymbol{H}$ is the Hessian of the loss and $\boldsymbol{G}$ can be interpreted as a Fisher Information Matrix, and show that the resulting scheme penalizes curvature in a principled (intrinsic and approximately coordinate-free) way, admits an interpretation as a kernel smoothing of the loss surface, and performs competitively on benchmark vision and NLP datasets.

## 1.1 Problem setting

In this work, we adopt two complementary perspectives and settings: a probabilistic one in which we view the neural network weights as random variables (Khan & Rue, 2021), and a point estimation setting in which the weights are fixed but unknown, and randomness arises from the composition of a deterministic neural network function and an output distribution (Martens, 2020). In the same way that ridge and lasso regularization can be derived from probabilistic models and correspond to certain priors (Gaussian and Laplacian, respectively) we derive results probabilistically and transfer them, by passing, via maximum a posteriori (MAP) estimation, to the point estimation setting on the underlying deterministic parameter space.

Our probabilistic setting is as follows: we are given a dataset $\mathcal{D} = \{(\boldsymbol{x_i}, \boldsymbol{y_i})_{i=1}^n\}$ consisting of $n$ independent input variables $\boldsymbol{x_i} \in \mathbb{R}^{d_x}$ with distribution $p(\boldsymbol{x})$, and corresponding targets (or labels) $\boldsymbol{y_i} \in \mathbb{R}^{d_y}$ with distribution $p(\boldsymbol{y}|\boldsymbol{x})$ and treat the parameters $\boldsymbol{w} \in \Theta \subseteq \mathbb{R}^p$ of a deep neural network (DNN) $f(\cdot, \boldsymbol{w})$ : $\mathbb{R}^{d_x} \to \mathbb{R}^{d_y}$ as random variables. Given a loss function $l(\boldsymbol{y_i}, f(\boldsymbol{x_i}, \boldsymbol{w}))$, our goal is to find a $\boldsymbol{w}^*$ that minimizes the expected loss: $\mathbb{E}_{p(\boldsymbol{x},\boldsymbol{y})}[l(\boldsymbol{y}, f(\boldsymbol{x}, \boldsymbol{w}))]$. Writing the finite-sample version of this expected loss as $L(\boldsymbol{w}) = \sum_{i=1}^n l(\boldsymbol{y_i}, f(\boldsymbol{x_i}, \boldsymbol{w}))$, we can form a generalized posterior distribution (Bissiri et al., 2016) $p(\boldsymbol{w}|\mathcal{D}) = p(\boldsymbol{w})\frac{1}{Z}\exp\{-L(\boldsymbol{w})\}$ over the weights with normalizer $Z$ and prior $p(\boldsymbol{w})$. This setting generalizes Bayesian inference (Knoblauch et al., 2019) to the case where the likelihood function is not specified as a conditional probability, but implicitly through an arbitrary loss function, and provides a coherent way to update prior beliefs over parameters to posterior beliefs (Bissiri et al., 2016) (Knoblauch et al., 2019). The generalized posterior coincides with the Bayesian posterior in the special case where the loss is given by a negative log-likelihood $L(\boldsymbol{w}) = -\frac{1}{n}\sum_{i=1}^n \log p(\boldsymbol{y_i}|\boldsymbol{x_i}, \boldsymbol{w})$.

We will seek to approximate the posterior using a family of distributions indexed by variational parameters, $q(\boldsymbol{w}) \sim \mathcal{N}(\boldsymbol{w}, \boldsymbol{\Sigma})$, by minimizing the KL-divergence between the true and approximate posteriors and casting the problem as an optimization over the variational parameters $\boldsymbol{\phi} = (\boldsymbol{\mu}, \boldsymbol{\Sigma})$. If we measure the dissimilarity between probability distributions using the KL divergence, the variational parameter space has a natural geometry induced by the Fisher Information Matrix (FIM):

$$
\begin{aligned}
\boldsymbol{F_\phi} &= \nabla^2_{\boldsymbol{\phi}'}\mathbb{D}_{KL}[q(\boldsymbol{w}; \boldsymbol{\phi}), q(\boldsymbol{w}; \boldsymbol{\phi}')]|_{\boldsymbol{\phi}'=\boldsymbol{\phi}} \\
&= \mathbb{E}_{\boldsymbol{w}\sim q}[\boldsymbol{\nabla_\phi}\log q(\boldsymbol{w}; \boldsymbol{\phi})^T\boldsymbol{\nabla_\phi}\log q(\boldsymbol{w}; \boldsymbol{\phi})]
\end{aligned}
\tag{1}
$$

which defines a Riemannian metric on $\boldsymbol{\phi}$. Since we have deliberately chosen a tractable variational distribution, we have an explicit expression for the FIM[4].

In our point estimation setting, we assume that we have an exponential family output probability distribution of the form $p(\boldsymbol{y}|\cdot)$ with natural parameters given by the output of a DNN $f(\boldsymbol{x}, \boldsymbol{w})$ so that $p(\boldsymbol{y}|f(\boldsymbol{x}, \boldsymbol{w}))$ defines a conditional distribution (Martens, 2020) (Kunstner et al., 2019). This setting covers the most common cases of multiclass classification with the cross-entropy loss[5] and least-squares regression. In this setting, the parameter space has as natural geometry induced by log-likelihood Fisher Information Matrix, or equivalently

---

[4]This is not, in general, the case for the log-likelihood Fisher Information Matrix

[5]where the softmax of neural network output is the vector of probabilities for each class (Martens, 2020)

the Hessian of the KL-divergence, which measures the dissimilarity between probability distributions in an intrinsic way, independently of parameterization:

$$\boldsymbol{F} = \nabla^2_{\boldsymbol{w}'} \mathbb{D}_{KL}[p(\boldsymbol{y}|\boldsymbol{x},\boldsymbol{w}), p(\boldsymbol{y}|\boldsymbol{x},\boldsymbol{w}')]|_{\boldsymbol{w}'=\boldsymbol{w}}$$
$$= \mathbb{E}_{\boldsymbol{y} \sim p(\boldsymbol{y}|\boldsymbol{x},\boldsymbol{w}), \boldsymbol{x} \sim p(\boldsymbol{x})}[(\nabla \log p(\boldsymbol{y}|\boldsymbol{x},\boldsymbol{w}))(\nabla \log p(\boldsymbol{y}|\boldsymbol{x},\boldsymbol{w}))^T] \tag{2}$$

As an infinitesimal form of the KL divergence, the log-likelihood FIM defines a metric tensor on the parameter manifold corresponding to the Fisher-Rao metric(Amari, 1998).

## 1.2 What is sharpness and why does it hurt generalization?

We first examine various notions of sharpness in the literature and then introduce the sharpness measure used in this paper.

### 1.2.1 The sharpness puzzle

Much of the literature on loss surface flatness and generalization has been concerned with measures of sharpness and flatness that depend on the particular choice of coordinate system. Notable examples include Keskar et al. (2016), who define $\epsilon$-sharpness as the maximum relative change in loss over a Euclidean norm-ball:

$$\max_{\|\boldsymbol{\Delta w}\|_2 \leq \epsilon} \frac{L(\boldsymbol{w} + \boldsymbol{\Delta w}) - L(\boldsymbol{w})}{1 + L(\boldsymbol{w})} \tag{3}$$

and the SAM algorithm (Foret et al., 2020), which uses a similar notion:

$$\max_{\|\boldsymbol{\Delta w}\|_2 \leq \epsilon} L(\boldsymbol{w} + \boldsymbol{\Delta w}) - L(\boldsymbol{w}) \tag{4}$$

Since, at such a local minimum of the loss, for a perturbation $\boldsymbol{\Delta w}$, we have:

$$L(\boldsymbol{w} + \boldsymbol{\Delta w}) - L(\boldsymbol{w}) = \boldsymbol{\Delta w}^T \boldsymbol{\nabla^2} L(\boldsymbol{w}) \boldsymbol{\Delta w} + O(\|\boldsymbol{\Delta w}\|^3) \tag{5}$$

both these measures are essentially equivalent to the spectral norm of the Hessian[6], which is not an invariant quantity. In particular, at any critical point which is a minimum with non-zero Hessian, there exists a reparameterization that leaves the underlying function of the data unchanged and which makes the spectral norm arbitrarily large (Dinh et al., 2017). More generally, there has been an extensive literature (Hochreiter & Schmidhuber, 1997) (Hinton & van Camp, 1993) attempting to characterize the loss-surface Hessian $\boldsymbol{\nabla^2} L(\boldsymbol{w})$ and to relate these characteristics to generalization. In many practically relevant cases, multiple minima are associated with zero (or close to zero) training error, and explicit or implicit regularization is needed to find solutions with the best generalization error.

Wei & Schwab (2020) show that given a degenerate valley in the loss surface, stochastic gradient descent (SGD) on average decreases the trace of the Hessian, which is suggestive of a connection between locally flat minima, overparameterization, and generalization. The parallel works of Sagun et al. (2017) and Chaudhari et al. (2016) examine the spectrum of the loss-function Hessian to characterize the landscape of the loss before and after optimization, and find that overparameterization is associated with the bulk of the Hessian spectrum lying close to zero and thus to highly degenerate minima. Observing that the clustering of eigenvalues around 0 corresponds to wide valleys in the loss surface, Chaudhari et al. (2016) propose an algorithm, Entropy-SGD, which has parallels with this work, and which explicitly introduces an "entropic term" which explicitly captures the width of the valley in the objective, resulting in a modified objective (to be maximized) (Dziugaite & Roy, 2017):

$$\log \mathbb{E}_{\boldsymbol{\epsilon} \sim \mathcal{N}(\boldsymbol{0}, \sigma^2 \boldsymbol{I})}[e^{-\tau L(\boldsymbol{w}+\boldsymbol{\epsilon})}] \tag{6}$$

Thus, the loss landscape is smoothed by applying a Gaussian convolution and the resulting minima are expected to be less sharp.

---

[6]To second order, $\max_{\|\boldsymbol{\Delta w}\|_2 \leq \epsilon} \frac{L(\boldsymbol{w}+\boldsymbol{\Delta w}) - L(\boldsymbol{w})}{1 + L(\boldsymbol{w})} \approx \epsilon \frac{||\boldsymbol{\nabla^2} L(\boldsymbol{w})||_2}{2(1 + L(\boldsymbol{w}))}$

Although these arguments have intuitive appeal, these notions of flatness are dependent on arbitrary choices of parameterization, which can, in general, be used to arbitrarily change the flatness of the loss surface without changing the underlying function of the input data (Dinh et al., 2017).

### 1.2.2 Our approach to sharpness

In this work we will implicitly measure sharpness as $\text{Tr}(\boldsymbol{G}^{-1}\boldsymbol{H})$ where $\boldsymbol{H}$ is the loss Hessian, and $\boldsymbol{G}$ can be interpreted as a Fisher Information Matrix, which, as a Riemannian metric tensor, defines a metric on the parameter manifold. At a critical point, $\text{Tr}(\boldsymbol{G}^{-1}\boldsymbol{H})$ is the Laplace-Beltrami operator[7] which generalizes the Laplacian to Riemannian manifolds (Lee, 2019) (Kristiadi et al., 2023), and defines an invariant, intrinsic quantity which, by analogy with $\text{Tr}(\boldsymbol{H})$ in Euclidean space, measures the average deviation from flatness, adjusting for the curvature of the manifold. Crucially, this notion is not an assumption but rather emerges naturally from the optimization of a generalized variational objective using the KL-metric. In particular, for a multivariate Gaussian variational approximation, the trace penalty corresponds to a smoothing of the loss surface using a kernel estimated online.

### 1.3 Penalizing sharpness: Sharpness-Aware Minimization

We first give an overview of SAM and its derivation, then highlight its strengths and weaknesses, and the relevant recent literature.

### 1.3.1 SAM overview

Despite the intuitive appeal and plausible justifications for flat solutions to be a goal of DNN optimization algorithms, there have been few unqualified practical successes in exploiting this connection to improve generalization performance. A notable exception is a recent algorithm, Sharpness Aware Minimization (SAM) (Foret et al., 2020), which seeks to improve generalization by optimizing a saddle-point problem of the form:

$$\min_{\boldsymbol{w}} \max_{\|\boldsymbol{\Delta w}\| \leq \rho} L(\boldsymbol{w} + \boldsymbol{\Delta w}) \tag{7}$$

An approximate solution to this problem is obtained by differentiating through the inner maximization, so that, given an approximate solution $\boldsymbol{\Delta w}^* := \rho \frac{\boldsymbol{\nabla} L(\boldsymbol{w}^k)}{\|\boldsymbol{\nabla} L(\boldsymbol{w}^k)\|_2}$ to the inner maximization (dual norm) problem:

$$\arg\max_{\|\boldsymbol{\Delta w}\| \leq \rho} L(\boldsymbol{w} + \boldsymbol{\Delta w}) \tag{8}$$

the gradient of the SAM objective is approximated as follows:

$$\boldsymbol{\nabla}_{\boldsymbol{w}} \left( \max_{\|\boldsymbol{\Delta w}\|_{FR} \leq \epsilon} L(\boldsymbol{w} + \boldsymbol{\Delta w}) \right) \approx \boldsymbol{\nabla}_{\boldsymbol{w}} L(\boldsymbol{w} + \boldsymbol{\Delta w}^*) \approx \boldsymbol{\nabla}_{\boldsymbol{w}} L(\boldsymbol{w})|_{\boldsymbol{w} + \boldsymbol{\Delta w}^*} \tag{9}$$

While the method has gained widespread attention, and state-of-the-art performance has been demonstrated on several benchmark datasets, it remains relatively poorly understood, and the motivation and connection to sharpness is questionable given that the Euclidean norm-ball isn't invariant to changes in coordinates. Given a 1-1 mapping $g : \Theta' \to \Theta$ we can reparameterize our DNN $f(\cdot, \boldsymbol{w})$ using the "pullback" $g^*(f)(\cdot, \boldsymbol{\nu}) := f(\cdot, g(\boldsymbol{\nu}))$ under which, crucially, the underlying prediction function $f(\cdot, \boldsymbol{w}) : \mathbb{R}^{d_x} \to \mathbb{R}^{d_y}$ (and therefore the loss) itself is invariant, since, for $\boldsymbol{\nu} = g^{-1}(\boldsymbol{w})$, we have $f(\cdot, \boldsymbol{w}) = f(\cdot, g(\boldsymbol{\nu}))$. Under this coordinate transformation, however, the Hessian at a critical point transforms as (Dinh et al., 2017):

$$\boldsymbol{\nabla^2} L(\boldsymbol{\nu}) = \boldsymbol{\nabla} g(\boldsymbol{\nu})^T \boldsymbol{\nabla^2} L \boldsymbol{\nabla} g(\boldsymbol{\nu}) \tag{10}$$

In particular, Dinh et al. (2017) explicitly show, using layer-wise transformations $T_\alpha : (\boldsymbol{w}_1, \boldsymbol{w}_2) \to (\alpha\boldsymbol{w}_1, \alpha^{-1}\boldsymbol{w}_2)$, that deep rectifier feedforward networks possess large numbers of symmetries which can be exploited to control sharpness without changing the network output. The existence of these symmetries

---

[7]Also known as the manifold Laplacian

in the loss function, under which the geometry of the local loss can be substantially modified (and in particular, the spectral norm and trace of the Hessian) means that the relationship between the local flatness of the loss landscape and generalization is a subtle one.

It is instructive to consider the PAC Bayes generalization bound that motivates SAM, the derivation of which starts from a PAC-Bayesian generalization bound (McAllester, 1999; Dziugaite & Roy, 2017):

**Theorem 1.** *For any distribution $\mathcal{D}$ and prior $p$ over the parameters $\boldsymbol{w}$, with probability $1 - \delta$ over the choice of the training set $\mathcal{S} \sim \mathcal{D}$, and for any posterior $q$ over the parameters:*

$$\mathbb{E}_q[L_{\mathcal{D}}(\boldsymbol{w})] \le \mathbb{E}_q[L_{\mathcal{S}}(\boldsymbol{w})] + \sqrt{\frac{\mathbb{D}_{KL}[q, p] + \log \frac{n}{\delta}}{2(n-1)}} \tag{11}$$

where the KL divergence:

$$\mathbb{D}_{KL}[q, p] = \mathbb{E}_{p(\boldsymbol{w})}\left[\log\left(\frac{q(\boldsymbol{w})}{p(\boldsymbol{w})}\right)\right] \tag{12}$$

defines a statistical distance $\mathbb{D}_{KL}[q, p]$ (though not a metric, as it's symmetric only to second order) on the space of probability distributions. Assuming an isotropic prior $p \sim N(\mathbf{0}, \sigma_p^2 \boldsymbol{I})$ for some $\sigma_p$, an isotropic posterior $q \sim N(\boldsymbol{w}, \sigma_q^2 \boldsymbol{I})$, so that $\mathbb{E}_q[L_{\mathcal{D}}(\boldsymbol{w})] = \mathbb{E}_{\boldsymbol{\epsilon} \sim \mathcal{N}(\mathbf{0}, \sigma_q^2 \boldsymbol{I})}[L_{\mathcal{D}}(\boldsymbol{w} + \boldsymbol{\epsilon})]$, applying the covering approach of Langford & Caruana (2001) to select the best (closest to $q$ in the sense of KL divergence) from a set of pre-defined data-independent prior distributions satisfying the PAC generalization bound, Foret et al. (2020) show that the bound in Theorem 1 can be written in the following form:

$$\mathbb{E}_{\boldsymbol{\epsilon} \sim \mathcal{N}(\mathbf{0}, \sigma_q^2 \boldsymbol{I})}[L_{\mathcal{D}}(\boldsymbol{w} + \boldsymbol{\epsilon})] \le \mathbb{E}_{\boldsymbol{\epsilon} \sim \mathcal{N}(\mathbf{0}, \sigma_q^2 \boldsymbol{I})}[L_{\mathcal{S}}(\boldsymbol{w} + \boldsymbol{\epsilon})] + g\left(\frac{\|\boldsymbol{w}\|_2^2}{\rho^2}\right) \tag{13}$$

(for a monotone function $g$). Then, crucially, one may apply a well-known tail-bound for a chi-square random variable to bound $\|\boldsymbol{\epsilon}\|_2$, thus bounding the expectation over $q$ (with probability $1 - 1/\sqrt{n}$) by the maximum value over a Euclidean norm-ball ball. This provides the following generalization bound:

**Theorem 2.** *For any $\rho > 0$ and any distribution $\mathcal{D}$, with probability $1 - \delta$ over the choice of the training set $\mathcal{S} \sim \mathcal{D}$,*

$$L_{\mathcal{D}}(\boldsymbol{w}) \le \max_{\|\boldsymbol{\epsilon}\|_2 \le \rho} L_{\mathcal{S}}(\boldsymbol{w} + \boldsymbol{\epsilon}) + g\left(\frac{\|\boldsymbol{w}\|_2^2}{\rho^2}\right) \tag{14}$$

where $\rho = \sigma\sqrt{k}\left(1 + \sqrt{\frac{\ln(n)}{k}}\right)$, $n = |\mathcal{S}|$, and $k$ is the number of parameters.

This bound justifies and motivates the SAM objective:

$$\max_{\|\boldsymbol{\Delta w}\| \le \rho} L(\boldsymbol{w} + \boldsymbol{\Delta w}) + \lambda\|\boldsymbol{w}\|_2^2 \tag{15}$$

and resulting algorithm. Although the bound in Theorem 2 suggests that the ridge penalty should vary with the radius of the perturbation, in practice (Foret et al., 2020) the penalty term is fixed (or simply set to zero) even when different perturbation radii are searched over. Subsequent refinements of SAM (Kim et al., 2022b) ignore the ridge penalty term altogether, and the choice of an optimal perturbation radius is what drives the success of the method. It is not clear, however, why this adversarial parameter-space perturbation should help generalization more than evaluating (and approximating) the expectation in the very bound which motivates the SAM procedure in the first place, which would lead instead to an objective (ignoring, for now, the ridge penalty term) of the following form:

$$\mathbb{E}_{\boldsymbol{\epsilon} \sim \mathcal{N}(\mathbf{0}, \sigma^2 \boldsymbol{I})}[L_{\mathcal{S}}(\boldsymbol{w} + \boldsymbol{\epsilon})] \tag{16}$$

Moreover, the worst-case adversarial perturbation used by SAM is likely to be noisier and is also naturally a significantly looser bound than the expectation-based bound.

### 1.3.2 SAM Strengths and weakness and related literature

SAM has shown great promise in some applications, particularly in its robustness to noise, where the performance gains are sometimes dramatic (Baek et al., 2024) (Foret et al., 2020). The method has also reinvigorated research on flatness-promoting regularizations. There are, however, numerous weaknesses and open questions, some of which have been addressed in the literature.

1. The Euclidean norm-ball based bound is not invariant to coordinate transformations, so that scale changes (such as occur, for example, when applying batch-normalization or weight-normalization) which have no effect on the output of the learned probability distribution, can nevertheless still result in arbitrary changes to the penalty. More generally, any geometric notion of loss surface flatness must be independent of arbitrary rescaling of the network parameters.

2. SAM performs poorly for large batch-sizes and the practical benefits of SAM are typically only seen for very small batch sizes (even though there is nothing in the theory or deviation to suggest this) (Andriushchenko & Flammarion, 2022).

3. SAM optimizes a loose upper-bound on an expectation in the generalization bound that motivates the method.

Several attempts have been made to address some of these issues. Kwon et al. (2021) focus on the inner maximization problem and propose ad hoc linear node-wise rescaling to mitigate the scale dependence of the method. Kim et al. (2022a) address the Euclidean norm-ball limitation by preconditioning the inner gradient step with an empirical inverse diagonal Fisher information matrix and demonstrate modest improvements over SAM on CIFAR-10 and CIFAR-100 datasets. Möllenhoff & Khan (2022) make the connection between SAM and Bayesian methods and show that SAM can be derived as an optimal relaxation of the Bayes objective, also demonstrating increased accuracy by improving the variance estimates and applying them to the inner gradient step.

## 2 G-TRACER

Motivated by these considerations, we introduce G-TRACER, a general regularization scheme which, given a base optimizer, consists of simply augmenting the loss with a term that penalizes the trace of the preconditioned Hessian $\boldsymbol{H}(\boldsymbol{w})$:

$$L^{\mathcal{G}}(\boldsymbol{w}) = L(\boldsymbol{w}) + \rho \text{Tr}(\boldsymbol{G}^{-1}\boldsymbol{H}(\boldsymbol{w})) \tag{17}$$

where the preconditioner is the inverse of an exponentially smoothed Fisher Information Matrix $\boldsymbol{F}$. When the base optimizer is chosen to be SGD, G-TRACER is given by following general update equations:

$$\boldsymbol{w} \leftarrow \boldsymbol{w} - \alpha \boldsymbol{\nabla}_{\boldsymbol{w}}[L(\boldsymbol{w}) + \rho \text{Tr}(\boldsymbol{G}^{-1}\boldsymbol{H}(\boldsymbol{w}))]$$
$$\boldsymbol{G} \leftarrow (1-\beta)\boldsymbol{G} + \beta \boldsymbol{F} \tag{18}$$

How is this related to SAM? Whereas SAM is derived from a loose bound on $\mathbb{E}_q[L(\boldsymbol{w})]$ by making the restrictive assumption that the posterior over the parameters is an isotropic Gaussian: $q(\boldsymbol{w}) \sim N(\boldsymbol{w}, \sigma_q^2 \boldsymbol{I})$ and by applying an (in general, loose) tail bound to the resulting expectation $\mathbb{E}_{\boldsymbol{\epsilon} \sim \mathcal{N}(\boldsymbol{0}, \sigma_q^2 \boldsymbol{I})}[L_{\mathcal{S}}(\boldsymbol{w} + \boldsymbol{\epsilon})]$, our regularization term is derived by forming a second-order approximation to $\mathbb{E}_q[L(\boldsymbol{w})] \approx L(\boldsymbol{\mu}) + \frac{1}{2}\text{Tr}(\boldsymbol{\Sigma}\boldsymbol{H})$ where $q(\boldsymbol{w}) \sim \mathcal{N}(\boldsymbol{w}, \boldsymbol{\Sigma})$, and the resulting regularization scheme is derived by performing natural gradient descent on the corresponding variational objective.

The effect is to modify the gradient from a pure descent direction $\boldsymbol{\nabla}_{\boldsymbol{w}} L(\boldsymbol{w})$, by a direction given by $\boldsymbol{\nabla}_{\boldsymbol{w}} \text{Tr}(\boldsymbol{G}^{-1}\boldsymbol{H}(\boldsymbol{w}))$, which encourages a reduction in the preconditioned Hessian trace. The preconditioner[8] $\boldsymbol{G}^{-1}$ can be viewed as an approximate inverse metric tensor that captures the geometry of the parameter space and the corresponding penalty term can be interpreted, in the neighbourhood of a critical point, as an approximate metric trace of the Hessian, or Laplace-Beltrami operator, which measures the difference

---

[8]Note that $\boldsymbol{G}^{-1}$ is a constant in the update equation for $\boldsymbol{w}$, whereas $\boldsymbol{H} \equiv \boldsymbol{H}(\boldsymbol{w})$

---

**Algorithm 1** SGD-TRACER
**Require:** $\alpha_t$: Stepsize
**Require:** $\beta$: Exponential smoothing constant for the online Fisher estimate
**Require:** $\rho$ : Flatness inducing penalty term
**Require:** $\delta$: Small positive constant
    Initialize $\boldsymbol{w}_0$, $\boldsymbol{f}_0$, $t = 0$
    **while** not converged **do**
        Sample batch $\mathcal{B} = \{(\boldsymbol{x}_1, \boldsymbol{y}_1), ...(\boldsymbol{x}_b, \boldsymbol{y}_b)\}$
        $\boldsymbol{w}_{t+1} = \boldsymbol{w}_t - \alpha_t \boldsymbol{\nabla}_{\boldsymbol{w}} \left[ L_{\mathcal{B}}(\boldsymbol{w}_t) + \rho \left( \boldsymbol{\nabla}_{\boldsymbol{w}} L_{\mathcal{B}}(\boldsymbol{w}_t) \right)^2 \cdot 1/(\boldsymbol{f}_t + \delta) \right]$
        $\boldsymbol{f}_{t+1} = (1 - \beta)\boldsymbol{f}_t + \beta \left( \boldsymbol{\nabla}_w L_{\mathcal{B}}(\boldsymbol{w}_t) \right)^2$
    **end while**

---

between the mean value of the loss over a geodesic ball (Lee, 2019) and the value at a minimum, and which thus promotes flatness. Crucially, flatness is promoted in a coordinate-free way, and the penalty is invariant to affine coordinate changes, such as the layer-wise scale transformations $T_\alpha : (\boldsymbol{w_1}, \boldsymbol{w_2}) \rightarrow (\alpha \boldsymbol{w_1}, \alpha^{-1} \boldsymbol{w_2})$ of Dinh et al. (2017), and more generally, at critical points, the penalty does not depend on the parameterization of the neural network $f$.

The general update equations (18) give rise to a number of possible practical regularization schemes. In this work, by approximating the Hessian and Fisher Information Matrix by the diagonal empirical Fisher Information Matrix, and replacing per-example squared gradients with squared minibatch gradients (the so-called gradient magnitude approximation (Bottou et al., 2016), we arrive at Algorithm 1, SGD-TRACER, where operations (other than the dot-product) are to be understood elementwise. In this simplified and restricted form, and as we show in detail in section 2.1, G-TRACER amounts to applying an adaptive natural-gradient L2-norm penalty since we have:

$$\rho\left(\boldsymbol{\nabla}_{\boldsymbol{w}} L(\boldsymbol{w_t})\right)^2 \cdot 1/\boldsymbol{f_t} = \rho || \tilde{\boldsymbol{F}}^{-1} \boldsymbol{\nabla}_{\boldsymbol{w}} L(\boldsymbol{w_t}) ||_2^2 \tag{19}$$

(where $\tilde{\boldsymbol{F}}$ is the diagonal matrix with diagonal given by $\sqrt{\boldsymbol{f_t}}$) which can be seen to be a preconditioned gradient norm, where the preconditioner is given by the square root of the inverse of the diagonal empirical Fisher (as used in adaptive optimizers such as Adam (Kingma & Ba, 2014)).

It is important to note that, in going from the general update equations 18 to a practical algorithm, many other choices are possible. The use of the diagonal (mean-field) empirical Fisher and the GM approximation are well established in the literature and are used in Adam (Kingma & Ba, 2014), Adagrad (Duchi et al., 2011) and RMSProp. Although these choices are straightforward to implement, scalable, widely used, and deliver competitive empirical results, a more principled approach with convergence guarantees (Kunstner et al., 2019) would be to use the Fisher information matrix (or equivalently, in most practically relevant settings, the Generalized Gauss-Newton (GGN) matrix). Recent advances in approximate second-order methods in optimization, notably Yao et al. (2020), suggest further avenues for improvement, and we leave investigations of alternatives, such as the smoothed (Hessian-free) Hessian diagonal sketch used in AdaHessian and KFAC (Martens & Grosse, 2015b) for future work. In general, there are likely to be significant gains in working with an approximate GGN matrix as well as in removing the gradient magnitude approximation, though we leave empirical investigations of these less straightforward alternatives for future work.

## 2.1 Connections with SAM and gradient norm penalization

The SGD-TRACER penalty can be written as

$$\rho\left(\boldsymbol{\nabla}_{\boldsymbol{w}} L(\boldsymbol{w_t})\right)^2 \cdot 1/\boldsymbol{f_t} = \rho || \tilde{\boldsymbol{F}}^{-1} \boldsymbol{\nabla}_{\boldsymbol{w}} L(\boldsymbol{w_t}) ||_2^2 \tag{20}$$

(where $\tilde{\boldsymbol{F}}$ is the diagonal matrix with diagonal given by $\sqrt{\boldsymbol{f_t}}$) The connection with sharpness aware optimization and its variants can be seen by taking the gradient of the augmented loss associated with the SGD-TRACER penalty:

$$L^{\mathcal{G}}(\boldsymbol{w_t}) = L(\boldsymbol{w_t}) + \rho || \tilde{\boldsymbol{F}}^{-1} \boldsymbol{\nabla}_{\boldsymbol{w}} L(\boldsymbol{w_t}) ||_2^2 \tag{21}$$

and choosing $\beta = 1$ in the update equation for $\boldsymbol{f_t}$ (corresponding to no smoothing) we have:

$$\boldsymbol{\nabla_w} L^{\mathcal{G}}(\boldsymbol{w_t}) = \boldsymbol{\nabla_w} L(\boldsymbol{w_t}) + 2\rho \boldsymbol{F}^{-1} \boldsymbol{H} \boldsymbol{\nabla_w} L(\boldsymbol{w_t}) \tag{22}$$

Writing the Hessian-vector product (to second order) as

$$\boldsymbol{H}\boldsymbol{\nabla_w} L(\boldsymbol{w_t}) \approx \frac{\boldsymbol{\nabla_w} L(\boldsymbol{w_t} + \delta\boldsymbol{\nabla_w} L(\boldsymbol{w_t})) - \boldsymbol{\nabla_w} L(\boldsymbol{w_t})}{\delta} \tag{23}$$

we get the following expression for the gradient of the penalized loss:

$$\begin{aligned}
\boldsymbol{\nabla_w} L^{\mathcal{G}}(\boldsymbol{w_t}) &= \boldsymbol{\nabla_w} L(\boldsymbol{w_t}) + 2\rho \boldsymbol{F}^{-1} \frac{\boldsymbol{\nabla_w} L(\boldsymbol{w_t} + \delta\boldsymbol{\nabla_w} L(\boldsymbol{w_t})) - \boldsymbol{\nabla_w} L(\boldsymbol{w_t})}{\delta} \\
&= (\boldsymbol{I} - \boldsymbol{\Gamma})\boldsymbol{\nabla_w} L(\boldsymbol{w_t}) + \boldsymbol{\Gamma}\boldsymbol{\nabla_w} L(\boldsymbol{w_t} + \delta\boldsymbol{\nabla_w} L(\boldsymbol{w_t}))
\end{aligned} \tag{24}$$

where $\boldsymbol{\Gamma} := 2\frac{\rho}{\delta}\boldsymbol{F}^{-1}$. Following the derivation of SAM and dropping higher-order terms, have the simplification: $\boldsymbol{\nabla_w} L(\boldsymbol{w_t} + \delta\boldsymbol{\nabla_w} L(\boldsymbol{w_t})) \approx \boldsymbol{\nabla_w} L(\boldsymbol{w})|_{\boldsymbol{w} = \boldsymbol{w_t} + \delta\boldsymbol{\nabla_w} L(\boldsymbol{w_t})}$, obtaining:

$$\boldsymbol{\nabla_w} L^{\mathcal{G}}(\boldsymbol{w_t}) = (\boldsymbol{I} - \boldsymbol{\Gamma})\boldsymbol{\nabla_w} L(\boldsymbol{w_t}) + \boldsymbol{\Gamma}\boldsymbol{\nabla_w} L(\boldsymbol{w})|_{\boldsymbol{w} = \boldsymbol{w_t} + \delta\boldsymbol{\nabla_w} L(\boldsymbol{w_t})} \tag{25}$$

The second term is proportional to $\boldsymbol{F}^{-1}\boldsymbol{\nabla_w} L(\boldsymbol{w})|_{\boldsymbol{w} = \boldsymbol{w_t} + \delta\boldsymbol{\nabla_w} L(\boldsymbol{w_t})}$ which is almost identical to the SAM gradient update (it is, in effect, a natural SAM gradient). In the special case that we choose $\boldsymbol{F} = \boldsymbol{I}$ and $\boldsymbol{\Gamma} = \boldsymbol{I}$ we recover effectively the same update as SAM:

$$\boldsymbol{\nabla_w} L^{\mathcal{G}}(\boldsymbol{w_t}) = \boldsymbol{\nabla_w} L(\boldsymbol{w})|_{\boldsymbol{w} = \boldsymbol{w_t} + \delta\boldsymbol{\nabla_w} L(\boldsymbol{w_t})} \tag{26}$$

which is exactly unnormalized SAM[9], as used in numerous theoretical works (Agarwala & Dauphin, 2023) and, commonly, by practitioners[10]. The corresponding augmented loss has the same solution set as the squared L2-norm penalty, though the optimization dynamics are, of course, different.

Thus, SAM can be seen as a special case of our more general scheme, corresponding to $\beta = 1$ (no smoothing), approximating an HVP, approximating the gradient of the resulting perturbed loss, and choosing as a preconditioner the identity matrix, so that the perturbation is not aligned with the natural geometry of the parameter space. Notably, at a critical point, the normalized SAM update isn't well defined since the perturbation is undefined. Whereas the normalized SAM perturbation radius is fixed, as our scheme approaches a critical point, the inverse Fisher has the effect of increasing the effective penalty in the directions where the gradient magnitudes are small.

Our scheme therefore also shows how gradient-norm penalties (Zhao et al., 2022) can be derived from probabilistic principles. In particular, natural-gradient norm penalization can thus be seen as a way to perform approximate variational inference.

Another complementary perspective on the relationship between a penalty of this form is that the supremum of the gradient norm of a real-valued locally Lipschitz-continuous function is the Lipschitz constant which controls the regularity of the function, and in particular bounds the change (in norm) of the output for a given change in the input (Zhao et al., 2022).

## 2.2 Derivation sketch

We first sketch the derivation of the general update equations (18) and then show how these lead to SGD-TRACER (1).

---

[9]Note that to recover the the scaled unit-norm perturbation $\boldsymbol{w_t} + \delta \frac{\boldsymbol{\nabla_w} L(\boldsymbol{w_t})}{||\boldsymbol{\nabla_w} L(\boldsymbol{w_t})||_2}$ from the original SAM paper would require a L2-norm gradient penalty: $\rho||\boldsymbol{\nabla_w} L(\boldsymbol{w_t})||_2$, see (Zhao et al., 2022) for a similar derivation

[10]see Andriushchenko & Flammarion (2022) for a proof of convergence of unnormalized SAM, who show that the normalized perturbation is not important for generalization

### 2.2.1 General update equations

Recall from Section 1.1 that we take a probabilistic approach and treat the parameters $\boldsymbol{w} \in \boldsymbol{\Theta} \subseteq \mathbb{R}^p$ of a neural network $f(\cdot, \boldsymbol{w}) : \mathbb{R}^{d_x} \to \mathbb{R}^{d_y}$ as random variables. Given a prior over the weights and a loss $L(w)$ we can form a (generalized) posterior distribution (Bissiri et al., 2016) $q(\boldsymbol{w}|\mathcal{D}) = p(\boldsymbol{w}) \frac{1}{Z} \exp\{-L(\boldsymbol{w})\}$ over the weights, with normalizer $Z$ and prior $p(\boldsymbol{w})$[11].

The posterior is given by the solution to the following generalized variational objective, (Knoblauch et al., 2019) (Bissiri et al., 2016) (the evidence lower bound (ELBO), which is related to, but more general than the objective from which SAM is derived) over the space of probability measures $\mathcal{P}(\boldsymbol{\Theta})$ on the parameter space $\Theta$:

$$q^*(\boldsymbol{w}) = \arg\min_{q \in \mathcal{P}(\boldsymbol{\Theta})} \left\{ \mathbb{E}_{q(\boldsymbol{w})}[L(\boldsymbol{w})] + \rho \mathbb{D}_{KL}[q, p] \right\} \tag{27}$$

where $\rho$ is a positive-valued parameter which controls the strength of regularization. We seek an approximate solution to equation 27 by assuming that the posterior $q(\boldsymbol{w})$ and prior $p(\boldsymbol{w})$ belong to multivariate Gaussian parametric families, so that the posterior has the form $q(\boldsymbol{w}) \sim \mathcal{N}(\boldsymbol{w}, \boldsymbol{\Sigma})$, and our optimization problem becomes:

$$\arg\min_{\boldsymbol{\phi}} \mathbb{E}_{q(\boldsymbol{w})}[L(\boldsymbol{w})] + \rho \mathbb{D}_{KL}[q, p] \tag{28}$$

where[12] $\boldsymbol{\phi} = (\boldsymbol{\mu}, \boldsymbol{\Sigma})$ and $\rho$ is a positive real-valued parameter.

Since we are optimizing in the space of probability distributions whose parameterizations can be changed without changing the underlying probability distribution, it is natural to perform gradient descent on our variational objective using Riemannian gradients corresponding to the Fisher-Rao metric[13] (also known as natural gradient descent (Amari, 1998)). This allows us to derive an algorithm that respects the intrinsic geometry of the parameter space, and thus derive an algorithm that seeks sharp minima in an approximately coordinate-independent way. The natural gradient operator in the variational parameter space $\boldsymbol{\phi} = (\boldsymbol{\mu}, \boldsymbol{\Sigma})$ is given by:

$$\tilde{\boldsymbol{\nabla}}_{\boldsymbol{\phi}} = \boldsymbol{F}_{\boldsymbol{\phi}}^{-1} \boldsymbol{\nabla}_{\boldsymbol{\phi}} \tag{29}$$

where the Fisher Information Matrix $\boldsymbol{F}_{\boldsymbol{\phi}}$ w.r.t. the variational parameters can be computed exactly and is given by (see Appendix A.1.2):

$$\boldsymbol{F}_{\boldsymbol{\phi}} = \begin{bmatrix} \boldsymbol{\Sigma}^{-1} & 0 \\ 0 & \frac{1}{2}(\boldsymbol{\Sigma}^{-1} \otimes \boldsymbol{\Sigma}^{-1}) \end{bmatrix}$$

We show in Appendix A.1.2 that, assuming an isotropic Gaussian prior, $p(\boldsymbol{w}) \sim \mathcal{N}(\boldsymbol{0}, \eta\boldsymbol{I})$, performing gradient descent on the objective 28 w.r.t. the natural gradient leads to the following iterative update equations(Khan & Rue, 2021) (Zhang et al., 2017):

$$\boldsymbol{\mu} \leftarrow \boldsymbol{\mu} - \alpha \boldsymbol{\Lambda}^{-1} \left( \mathbb{E}_q[\boldsymbol{\nabla}_{\boldsymbol{w}} L(\boldsymbol{w})] + \frac{\rho}{\eta} \boldsymbol{w} \right)$$
$$\boldsymbol{\Lambda} \leftarrow (1 - \beta)\boldsymbol{\Lambda} + \beta \left( \frac{\mathbb{E}_q[\boldsymbol{\nabla}_{\boldsymbol{w}}^2 L(\boldsymbol{w})]}{\rho} + \eta^{-1}\boldsymbol{I} \right) \tag{30}$$

where $\alpha$ and $\beta$ are the learning rates for the mean and precision updates, respectively, and $\boldsymbol{\Lambda} := \boldsymbol{\Sigma}^{-1}$ is the precision matrix. Approximating the expectations to second order and further simplifying[14] leads to the update equations (see Appendix A.1.2 for a detailed derivation):

$$\boldsymbol{\mu} \leftarrow \boldsymbol{\mu} - \alpha \boldsymbol{\nabla}_{\boldsymbol{\mu}}[L(\boldsymbol{\mu}) + \rho \text{Tr}(\boldsymbol{\Lambda}^{-1} \boldsymbol{H}(\boldsymbol{\mu}))]$$
$$\boldsymbol{\Lambda} \leftarrow (1 - \beta)\boldsymbol{\Lambda} + \beta \boldsymbol{H} \tag{31}$$

---

[11]This coincides with the Bayesian posterior in the special case that the loss is the negative log-likelihood $L(\boldsymbol{w}) = -\frac{1}{n}\sum_{i=1}^n \log p(\boldsymbol{y_i}|\boldsymbol{x_i}, \boldsymbol{w})$

[12]We will write, to keep the notation as light as possible, the set of variational parameters as $\boldsymbol{\phi} = (\boldsymbol{\mu}, \boldsymbol{\Sigma})$. Depending on the context, e.g. when we write the gradient $\boldsymbol{\nabla}_{\boldsymbol{\phi}}$, we will take this to mean $\phi = \begin{bmatrix} \boldsymbol{\mu} \\ \text{vec}(\boldsymbol{\Sigma}) \end{bmatrix}$

[13]This can be shown to be steepest descent in the KL-metric (Martens, 2020)

[14]We drop the gradient preconditioner in the update equation for $\mu$ for simplicity of exposition, since our focus here is on the regularizer

where $\boldsymbol{H} = \boldsymbol{\nabla}_{\boldsymbol{\mu}}^{2} L(\boldsymbol{\mu})$ is the Hessian.

As we will see in the full derivation of the method in Appendix A.1.2, the recursion for $\boldsymbol{\Lambda}$ involving the loss Hessian is an update equation for the precision[15], which is exactly the Fisher Information Matrix $\boldsymbol{F}_{\boldsymbol{\mu}}$ of the variational distribution $q$ viewed as a function of $\boldsymbol{\mu}$:

$$\boldsymbol{F}_{\boldsymbol{\mu}} = \mathbb{E}_{\boldsymbol{w} \sim q}[\boldsymbol{\nabla}_{\boldsymbol{\mu}} \log q(\boldsymbol{w}; \boldsymbol{\mu})^{T} \boldsymbol{\nabla}_{\boldsymbol{\mu}} \log q(\boldsymbol{w}; \boldsymbol{\mu})] = \mathbb{E}_{\boldsymbol{w} \sim q}[-\boldsymbol{\nabla}_{\boldsymbol{\mu}}^{2} \log q(\boldsymbol{w}; \boldsymbol{\mu})^{T}] \tag{32}$$

and which defines a Riemannian metric on $\boldsymbol{\mu}$.

### 2.3 MAP estimation and general update equations

So far we have worked in variational parameter space, and we now wish to pass to the underlying parameter space. We do so by taking the maximum a posteriori (MAP) estimate of the multivariate Gaussian posterior, which allows us to identify $\boldsymbol{\mu}$ and $\boldsymbol{w}$. We are thus summarizing the multivariate posterior by its mean (and modal) value. Since we have passed to the underlying parameter space and the update equation for $\boldsymbol{\Lambda}$ is used only to compute the preconditioner in the update equation for the mean and not to maintain a full distribution over the parameters, we denote this smoothed Hessian by $\boldsymbol{G}$. Our update equation now takes the form:

$$\begin{aligned} \boldsymbol{w} &\leftarrow \boldsymbol{w} - \alpha \boldsymbol{\nabla}_{\boldsymbol{w}}[L(\boldsymbol{w}) + \rho \mathrm{Tr}(\boldsymbol{G}^{-1} \boldsymbol{H}(\boldsymbol{w}))] \\ \boldsymbol{G} &\leftarrow (1 - \beta)\boldsymbol{G} + \beta \boldsymbol{H} \end{aligned} \tag{33}$$

Finally, we approximate the Hessian $\boldsymbol{H}$ in the update equation for $\boldsymbol{G}$ (equation 33) by the log-likelihood Fisher Information Matrix (FIM),

$$\begin{aligned} \boldsymbol{F} &= \mathbb{E}_{\boldsymbol{y} \sim p(\boldsymbol{y}|\boldsymbol{x}, \boldsymbol{w}), \boldsymbol{x} \sim p(\boldsymbol{x})}[(\boldsymbol{\nabla} \log p(\boldsymbol{y}|\boldsymbol{x}, \boldsymbol{w}))(\boldsymbol{\nabla} \log p(\boldsymbol{y}|\boldsymbol{x}, \boldsymbol{w}))^{T}] \\ &= \mathbb{E}_{\boldsymbol{y} \sim p(\boldsymbol{y}|\boldsymbol{x}, \boldsymbol{w}), \boldsymbol{x} \sim p(\boldsymbol{x})}[-\boldsymbol{\nabla}_{\boldsymbol{w}}^{2} \log p(\boldsymbol{y}|\boldsymbol{x}, \boldsymbol{w})] \end{aligned} \tag{34}$$

which, as the expectation of the negative log-likelihood Hessian under the model's output distribution[16] (Amari, 1998), is known to provide a better local quadratic approximation to $L(\boldsymbol{w})$ than the Hessian $H$ (Martens & Sutskever, 2012) when optimizing non-convex objectives. In particular:

$$M(\boldsymbol{\Delta}_{\boldsymbol{w}}) = \frac{1}{2} \boldsymbol{\Delta}_{\boldsymbol{w}}^{T} \boldsymbol{F} \boldsymbol{\Delta}_{\boldsymbol{w}} + \boldsymbol{\nabla}_{\boldsymbol{w}} L(\boldsymbol{w})^{T} \boldsymbol{\Delta}_{\boldsymbol{w}} + L(\boldsymbol{w}) \tag{35}$$

can be viewed as a convex approximation to the second-order Taylor series of $L(\boldsymbol{w} + \boldsymbol{\Delta}_{\boldsymbol{w}})$ for which the minimizer is the negative natural gradient $-\boldsymbol{F}^{-1} \boldsymbol{\nabla}_{\boldsymbol{w}} L(\boldsymbol{w})$ (Martens, 2020).

We therefore substitute the $\boldsymbol{F}$ for the Hessian $\boldsymbol{H}$ and, in addition to the advantages of positive definiteness and strong empirical performance, this is a natural identification when the loss is given by the log-loss $l(\boldsymbol{y}, f(\boldsymbol{x}, \boldsymbol{w})) = -\log p(\boldsymbol{y}|\boldsymbol{x}, \boldsymbol{w})$[17] where the log-likelihood FIM converges to the Hessian as training error goes to zero, and where it is equivalent to the Generalized Gauss-Newton matrix (Kunstner et al., 2019) (Martens, 2020). In particular, by Proposition 1 of Kunstner et al. (2019) (see also Martens (2020)), we have that, under mild smoothness assumptions (in particular, assuming that the coordinate functions of the DNN are $\beta$-smooth), and assuming $p(\boldsymbol{y}|f(\boldsymbol{x}, \boldsymbol{w}))$ is an exponential family distribution[18]:

$$||\boldsymbol{\nabla}^{2} L(\boldsymbol{w}) - \mathbf{F}||_{2}^{2} < r(\boldsymbol{w})\beta \tag{36}$$

where the residual $r(\boldsymbol{w}) = \sum_{n=1}^{N} ||\boldsymbol{\nabla}_{\boldsymbol{f}} \log p(\boldsymbol{y_n}|f(\boldsymbol{x_n}, \boldsymbol{w}))||$ goes to zero as the training error goes to zero.

Crucially, the log-likelihood FIM is also the Hessian of the KL divergence, which measures the dissimilarity between probability distributions in an intrinsic way, independently of parameterization. As an infinitesimal

---

[15]We explain later why this is a natural choice

[16]Whereas $H$ is the expected Hessian under the data distribution

[17]And for the most practically relevant losses (which are the ones we consider here): cross-entropy (classification), and squared error (regression), corresponding to exponential family output distributions with natural parameters given by the output function $f(\boldsymbol{x}, \boldsymbol{w})$ (Kunstner et al., 2019) (Martens, 2020)

[18]see (Kunstner et al., 2019) for further details, including how to define the output distribution and loss so the the GGN and FIM are equivalent

form of the KL divergence, it defines a metric tensor on the parameter manifold corresponding to the Fisher-Rao metric(Amari, 1998). The use of the Fisher approximation to the Hessian therefore connects in a natural way the Riemannian metric on the space of variational parameters $\boldsymbol{F_\mu}$ and the approximate Riemannian metric $\boldsymbol{F}$ in the underlying parameter space of weights $\boldsymbol{\Theta}$.

We show in Appendix A.1.2 that the penalty term $\rho\mathrm{Tr}(\boldsymbol{G^{-1}H})$ is invariant to affine coordinate transformations. In the neighborhood of local minima we have approximate invariance to general smooth coordinate changes, since $\boldsymbol{G^{-1}}$ is an approximation to the inverse metric on the parameter space $\boldsymbol{\Theta}$ which, as a (0,2) tensor transforms in the same way as the Hessian at a critical point, where the Christoffel symbols vanish (Lee, 2019). The penalty can be interpreted, in the neighborhood of local minima, as a coordinate-independent measure of curvature that correctly accounts for the geometry of the underlying manifold $L(\boldsymbol{w})$.

Substituting-in the FIM approximation of the Hessian (see Appendix A.1.2 for details, and a discussion of alternatives) we arrive at the general update equations (18):

$$\boldsymbol{w} \leftarrow \boldsymbol{w} - \alpha\boldsymbol{\nabla_w}[L(\boldsymbol{w}) + \rho\mathrm{Tr}(\boldsymbol{G^{-1}H(w)})]$$
$$\boldsymbol{G} \leftarrow (1-\beta)\boldsymbol{G} + \beta\boldsymbol{F} \tag{37}$$

### 2.3.1 SGD-TRACER

The Empirical Fisher Information Matrix, given by

$$\tilde{\boldsymbol{F}} := \sum_{i=1}^{n} \boldsymbol{\nabla_w} \log p(\boldsymbol{y_i}|\boldsymbol{x_i}, \boldsymbol{w})^T \boldsymbol{\nabla_w} \log p(\boldsymbol{y_i}|\boldsymbol{x_i}, \boldsymbol{w}) \tag{38}$$

which replaces the FIM's expectation over the model's output distribution by an expectation over the training data, despite lacking the same convergence guarantees, performs competitively in many settings (Kunstner et al., 2019). We find in our experiments that the empirical Fisher performs competitively with the MC approximation to the FIM/GGN (Khan et al., 2018; Kingma & Ba, 2014) and has the advantage of being straightforward and cheap to compute from already computed gradients (in the case of applying a G-TRACER regularizer to Adam, the smooth squared gradients are already computed and maintained for use as a preconditioner). Given the conceptual and computational simplicity of this approach we substitute the empirical Fisher for the Fisher Information Matrix, as well as for the Hessian in the update equation for the mean, giving the update equations:

$$\boldsymbol{w} \leftarrow \boldsymbol{w} - \alpha\boldsymbol{\nabla_w}[L(\boldsymbol{w}) + \rho\mathrm{Tr}(\boldsymbol{G^{-1}}\tilde{\boldsymbol{F}})]$$
$$\boldsymbol{G} \leftarrow (1-\beta)\boldsymbol{G} + \beta\tilde{\boldsymbol{F}} \tag{39}$$

Recent advances in approximate second-order methods in optimization, notably Yao et al. (2020), suggest avenues for improvement, and we leave investigations of alternatives, such as the smoothed (Hessian-free) Hessian diagonal sketch used in AdaHessian, for future work.

We now make two simplifications. First, we use a mean-field approximation, representing the empirical FIM by its diagonal:

$$\tilde{\boldsymbol{F}} \approx \frac{1}{n}\sum_{i=1}^{n} \boldsymbol{\nabla_w} \log p(\boldsymbol{y_i}|\boldsymbol{x_i}, \boldsymbol{w})^2 \tag{40}$$

The inverse of the diagonal empirical Fisher, as an approximation to the inverse Fisher Information Matrix, is used as a gradient preconditioner by the Adam (Kingma & Ba, 2014) and Adagrad (Duchi et al., 2011) optimizers. Although many other approaches are possible here, of varying degrees of sophistication and complexity (see KFAC (Martens & Grosse, 2015a), for example), we find that this simple, cheap to compute, and scalable approach works extremely well.

Secondly, as most current deep learning frameworks do not straightforwardly support access to per-example gradients, which can in principle be achieved with negligible additional cost (see, for example, BackPACK (Dangel et al., 2020) second-order Pytorch extensions), for simplicity and efficiency, we use the gradient

magnitude (GM) approximation (Bottou et al., 2016), as used in standard optimizers Adam and RMSprop, replacing the sum of squared gradients with the square of summed gradients:

$$\frac{1}{n}\sum_{i=1}^{n}[\boldsymbol{\nabla_w}\log p(\boldsymbol{y_i}|\boldsymbol{x_i},\boldsymbol{w})]^2 \approx \left[\frac{1}{n}\sum_{i=1}^{n}\boldsymbol{\nabla_w}\log p(\boldsymbol{y_i}|\boldsymbol{x_i},\boldsymbol{w})\right]^2 \tag{41}$$

and we write the resulting FIM diagonal as $\boldsymbol{\nabla_w}L(\boldsymbol{w})^2$. Finally, it is standard practice to (Martens, 2020) to add Tikhonov regularization or damping via a small positive real constant $\delta$ when using 2nd-order optimization methods, and, passing to mini-batches, we end up with Algorithm 1 (with elementwise ops):

$$\boldsymbol{w}_{t+1} = \boldsymbol{w}_t - \alpha_t \boldsymbol{\nabla_w}\left[L_{\mathcal{B}}(\boldsymbol{w}_t) + \rho\left(\boldsymbol{\nabla_w}L_{\mathcal{B}}(\boldsymbol{w}_t)\right)^2 \cdot 1/(\boldsymbol{f}_t + \delta)\right]$$
$$\boldsymbol{f}_{t+1} = (1-\beta)\boldsymbol{f}_t + \beta\left(\boldsymbol{\nabla_w}L_{\mathcal{B}}(\boldsymbol{w}_t)\right)^2 \tag{42}$$

in which the usual stochastic gradient update is modified with a term which penalizes the trace of the ratio of the the empirical FIM and an exponentially weighted average the of the empirical FIM. By augmenting the loss with a TRACER term and maintaining a smoothed squared-gradient estimate, in principle, any optimization scheme can be modified in the same way. In our experiments, we use SGD with momentum for vision tasks and Adam-TRACER (Adam with a G-TRACER penalty) for NLP tasks, based on standard practice in each problem domain.

## 3 How does G-TRACER penalize sharpness?

We first examine, in a simplified setting, the interplay between the penalty parameter $\rho$ and the variance of the perturbation over which the loss is smoothed by convolution with a Gaussian kernel. We then show that, when the expectation (or convolution) is approximated to second order, the result has a direct correspondence with the Laplace-Beltrami operator, which establishes a rigorous link to flatness. We then show, by passing to the underlying weight space, how the final form of the regularizer also admits an interpretation as a flatness inducing penalty in the log-likelihood FIM-induced geometry of the underlying parameter space. Finally, we examine links between gradient norm penalties, SAM and SGD-TRACER.

### 3.1 $\rho$ determines generalized variance of the Gaussian kernel smoothing

Starting with equation 28 and ignoring for simplicity the contribution from the prior term, which would correspond to a ridge-regularization term under the assumption $p(\boldsymbol{w}) \sim \mathcal{N}(\boldsymbol{0}, \sigma_p \boldsymbol{I})$, we have the following objective, which we seek to minimize over $\boldsymbol{w}$:

$$\mathbb{E}_{q(\boldsymbol{w})}[L(\boldsymbol{w})] - \rho\mathcal{H}(q) \tag{43}$$

where $\mathcal{H}(q) = \mathbb{E}_{q(\boldsymbol{w})}[-\log q(\boldsymbol{w})]$ is the entropy of $q$. For the choice $q(\boldsymbol{w}) \sim \mathcal{N}(\boldsymbol{\mu}, \sigma^2 \boldsymbol{I})$, the optimization problem associated with the variational objective becomes (absorbing some constants into $\rho$):

$$\arg\min_q \mathbb{E}_q[L(\boldsymbol{w})] - \rho\mathcal{H}(q) = \arg\min_{\boldsymbol{w},\sigma^2} \mathbb{E}_q[L(\boldsymbol{w})] + \rho\log\frac{1}{\sigma^2} \tag{44}$$

so that we can see that $\rho$ determines the variance of Gaussian perturbation over which the loss is averaged. More generally, choosing $q \sim \mathcal{N}(\boldsymbol{w}, \boldsymbol{\Sigma})$ leads to the following variational objective:

$$\arg\min_q \mathbb{E}_q[L(\boldsymbol{w})] - \rho\mathcal{H}(q) = \arg\min_{\boldsymbol{w},\boldsymbol{\Sigma}} \mathbb{E}_q[L(\boldsymbol{w})] + \rho\log\frac{1}{\det(\boldsymbol{\Sigma})} \tag{45}$$

so that large values of $\rho$ will correspond to distributions with larger volume, since for $\boldsymbol{x} \sim \mathcal{N}(\boldsymbol{0}, \boldsymbol{\Sigma})$, $\boldsymbol{x}$ lies within the ellipsoid $\boldsymbol{x}^T\boldsymbol{\Sigma}^{-1}\boldsymbol{x} = \chi^2(\alpha)$ with probability $1 - \alpha$, with the volume of the ellipsoid proportional to $\det(\boldsymbol{\Sigma})^{\frac{1}{2}}$ (Anderson, 2003). The regularization parameter $\rho$ thus controls the generalized variance $\det(\boldsymbol{\Sigma})$ of the Gaussian kernel which is used to smooth the loss when calculating the expectation $\mathbb{E}_{q(\boldsymbol{w})}[L(\boldsymbol{w})]$.

### 3.2 $\mathrm{Tr}(\mathbf{G}^{-1}\mathbf{H})$ regularization is curvature regularization and thus promotes flatness

We show how G-TRACER promotes flatness from three complementary perspectives, corresponding to the three forms of the update equations, in decreasing order of generality. First, we consider the variational update equations (31), corresponding to the variational parameter space, in which we view the neural network parameters as random variables. Second, we examine the parameter-space general update equations (18) relating to the underlying weight space, in which the randomness arises from the output distribution of the neural network. Third, we highlight the connections between gradient norm regularization, SAM, and the final simplified SGD-TRACER formulation (1) from a parameter-space perspective.

#### 3.2.1 Variational parameter space sharpness

We first examine curvature regularization from the perspective of the variational parameter space. In this probabilistic setting, the parameters of the neural network are random variables. The geometric trace penalty arises by modeling the posterior over the parameters as a multivariate Gaussian: $q(\boldsymbol{w}) \sim \mathcal{N}(\boldsymbol{\mu}, \boldsymbol{\Sigma})$, and then performing natural gradient descent on the variational objective (55) resulting in the update equations 31:

$$\boldsymbol{\mu} \leftarrow \boldsymbol{\mu} - \alpha\boldsymbol{\nabla}_{\boldsymbol{\mu}}[L(\boldsymbol{\mu}) + \rho\mathrm{Tr}(\boldsymbol{\Lambda}^{-1}\boldsymbol{H}(\boldsymbol{\mu}))]$$
$$\boldsymbol{\Lambda} \leftarrow (1-\beta)\boldsymbol{\Lambda} + \beta\boldsymbol{H} \tag{46}$$

The Fisher Information Matrix $\boldsymbol{F_\mu}$ of $q$, viewed as a function of the variational mean parameter[19] $\boldsymbol{\mu}$, can be computed exactly, and is given by:

$$\boldsymbol{F_\mu} = \mathbb{E}_q\left[-\boldsymbol{\nabla}_{\boldsymbol{\mu}}^2 \log q\right] = \boldsymbol{\Lambda} \tag{47}$$

(see Appendix A.1.2 for details). Thus the penalty term $\mathrm{Tr}(\boldsymbol{\Lambda}^{-1}\boldsymbol{H}(\boldsymbol{\mu})) \propto \mathrm{Tr}(\boldsymbol{F}_{\boldsymbol{\mu}}^{-1}\boldsymbol{H}(\boldsymbol{\mu})) = \mathrm{Tr}_{\boldsymbol{F_\mu}}(\boldsymbol{H})$, where $\mathrm{Tr}_{\boldsymbol{F_\mu}}$ is the metric trace[20] (Lee, 2019) of the Hessian, and which, at a critical point, is exactly the Laplacian $\boldsymbol{\Delta}$ (also known as the Laplace-Beltrami operator (Lee, 2019) Kristiadi et al. (2023)), a fundamental operator in differential geometry and analysis, which measures curvature in an intrinsic, coordinate independent way, correcting for the underlying geometry of the manifold[21]. In the neighborhood of a local minimum $\boldsymbol{\mu^*}$ of $L$ (in particular), it can be interpreted as the difference between the mean value of $L$ over a (geodesic) ball centered at $\boldsymbol{\mu^*}$ and $L(\boldsymbol{\mu^*})$, due to the following mean-value property for smooth functions over geodesic balls $B_r(\boldsymbol{\mu})$ (Gray & Willmore, 1982) (Loustau, 2015):

$$\frac{1}{\mathrm{vol}(B_r(\boldsymbol{\mu^*}))}\int_{B_r(\boldsymbol{\mu^*})} L(\boldsymbol{\mu})d\boldsymbol{V} - L(\boldsymbol{\mu^*}) = \frac{\boldsymbol{\Delta}L(\boldsymbol{\mu^*})}{2(n+2)}r^2 + \mathcal{O}(r^4) \tag{48}$$

so that we have the following asymptotic expression for the value of the Laplacian at $\boldsymbol{\mu^*}$:

$$\boldsymbol{\Delta}L(\boldsymbol{\mu^*}) = \lim_{r\to 0}\frac{2(n+2)}{r^2}\frac{1}{\mathrm{vol}(B_r(\boldsymbol{\mu^*}))}\int_{B_r(\boldsymbol{\mu^*})} L(\boldsymbol{\mu}) - u(\boldsymbol{\mu^*})d\boldsymbol{V} \tag{49}$$

Since, at a minimum $\boldsymbol{\mu^*}$ of $L(\boldsymbol{\mu})$, $\mathrm{Tr}(\boldsymbol{F}_{\boldsymbol{\mu^*}}^{-1}\boldsymbol{H}) = \boldsymbol{\Delta}L(\boldsymbol{\mu^*}) \geq 0$, penalizing $\mathrm{Tr}(\boldsymbol{F}_{\boldsymbol{\mu}}^{-1}\boldsymbol{H})$ has the effect of forcing the values of $L(\boldsymbol{\mu})$ in a neighborhood of a minimum to be closer to the value at the minimum.

#### 3.2.2 Underlying parameter-space sharpness

From the perspective of the underlying parameter space $\boldsymbol{\Theta}$ (the weight space), the general update equations (equation 18) consist of the G-TRACER penalty $\rho\mathrm{Tr}(\boldsymbol{G}^{-1}\boldsymbol{H})$ (which is affine invariant, as well as invariant

---

[19]$\boldsymbol{\Sigma}$ is a constant in the update equation for $\boldsymbol{\mu}$

[20]The metric induces canonical or musical isomorphisms $\sharp$ and $\flat$ between the tangent and cotangent bundles. The metric trace of a symmetric 2-tensor $\boldsymbol{H}$ is $\mathrm{Tr}_{\boldsymbol{F_\mu}}\boldsymbol{H} = \mathrm{Tr}\boldsymbol{H}^\sharp$

[21]An alternative viewpoint, complementary to the one we take here, is an extrinsic one, where we consider the loss surface in graph coordinates as a hypersurface embedded in ambient Euclidean space $\mathbb{R}^{p+1}$: $\{(\boldsymbol{w}, L(\boldsymbol{w})), \boldsymbol{w} \in \mathbb{R}^{p+1} : \boldsymbol{w} \in \Theta\}$. In Euclidean space, the Hessian is the matrix of the shape operator (or the second fundamental form (Lee, 2019)). The eigenvalues of the Hessian correspond to the principal curvatures, and the mean curvature is the mean of the principal curvatures (or equivalently, the Hessian trace). The Euclidean metric on $\mathbb{R}^{p+1}$ then induces a pullback metric on the embedded submanifold, $\boldsymbol{G}$, and at a critical point, the matrix of the shape operator in a Riemannian manifold is given by $\boldsymbol{G}^{-1}\boldsymbol{H}$ with corresponding mean curvature $\mathrm{Tr}(\boldsymbol{G}^{-1}\boldsymbol{H})$ (Kristiadi et al., 2023).

to all diffeomorphic coordinate transformations at critical points), where $\boldsymbol{G}$ is an exponentially smoothed log-likelihood FIM. $\boldsymbol{G}$ captures the local geometry of the loss surface $L(\boldsymbol{w})$ and, in the neighborhood of critical points, the G-TRACER penalty, as an approximate metric trace w.r.t. $\boldsymbol{G}$, can be interpreted as a measure of sharpness on the weight space. The choice $\beta = 1$ in the update equation (corresponding to no smoothing) for $\boldsymbol{G}$ corresponds to the case where $\rho \text{Tr}(\boldsymbol{G}^{-1}\boldsymbol{H})$ is exactly log-likelihood FIM-preconditioned Hessian trace.

### 3.3 Preconditioned gradient norm penalty

The final simplified form of the udpate equations, SGD-TRACER, amounts to an adaptive preconditioned gradient norm penalty that can be seen as a generalization of SAM. The penalty has the form:

$$\rho \big( \boldsymbol{\nabla}_{\boldsymbol{w}} L(\boldsymbol{w_t}) \big)^2 \cdot 1/\boldsymbol{f_t} = \rho || \tilde{\boldsymbol{F}}^{-1} \boldsymbol{\nabla}_{\boldsymbol{w}} L(\boldsymbol{w_t}) ||_2^2 \tag{50}$$

(where $\tilde{\boldsymbol{F}}$ is the diagonal matrix with diagonal given by $\sqrt{\boldsymbol{f_t}}$) and the resulting gradient update, after a number of simplifications and approximations (see section 2.1), has the form:

$$\boldsymbol{\nabla}_{\boldsymbol{w}} L^{\mathcal{G}}(\boldsymbol{w_t}) = (\boldsymbol{I} - \boldsymbol{\Gamma}) \boldsymbol{\nabla}_{\boldsymbol{w}} L(\boldsymbol{w_t}) + \boldsymbol{\Gamma} \boldsymbol{\nabla}_{\boldsymbol{w}} L(\boldsymbol{w})|_{\boldsymbol{w} = \boldsymbol{w_t} + \delta \boldsymbol{\nabla}_{\boldsymbol{w}} L(\boldsymbol{w_t})} \tag{51}$$

where $\boldsymbol{\Gamma} := 2\frac{\rho}{\delta} \boldsymbol{F}^{-1}$ and which strongly resembles a natural-gradient SAM update. As a gradient norm penalty, the approach can also be seen, from a functional analytic perspective, as controlling the Lipschitz constant and thus promoting regularity. Our method thus shows how natural gradient norm penalties can be derived from general probabilistic principles.

## 4 Results

While the original SAM paper (Foret et al., 2020) and subsequent papers Kwon et al. (2021) (Kim et al., 2022a) largely focus on standard benchmark problems and show marginal improvements in many settings, the most striking and practically relevant improvements concern the performance gains in the more challenging noisy-label settings. The CIFAR-10 and CIFAR-100 benchmarks are extremely well understood, and good training schedules, data augmentations, and architecture choices have all been found over an extremely large number of trials run by the community over many years. The effect size of augmentations is often large (10% accuracy gains, or more) compared to post-augmentation gains exhibited by SAM (typically on the order of 1%).

Given our focus on delivering material performance gains in challenging settings, we examine the performance of our algorithm on especially challenging variants of standard benchmarks as they are a good model for the kinds of real-world applications that most require general-purpose regularizations. For example, we apply our method to noisy variants (with up to 50% label noise) of CIFAR-100, with and without data augmentations, first using a standard ResNet architecture and then on a vision transformer architecture (ViT). The ViT is trained from scratch (no pre-training), which is extremely challenging, since the convolution's inductive bias of spatial locality is lost in moving to a transformer architecture. We are thus using variants on standard benchmarks as a model for general settings, such as financial time-series forecasting, in which low effective sample size, extremely low signal-noise ratio, extreme nonstationarity, and a general lack of symmetries (financial time series exhibit neither up-down symmetry nor time-reversal symmetry) which give rise to data augmentations, all contrive to make generalization very difficult.

Finally, we show encouraging results on challenging subtasks from one of the most challenging NLP benchmarks, SuperGlue, using the BERT transformer architecture.

### 4.1 Vision: CIFAR-100, challenging variants

We first examine a variant on a standard benchmark in computer vision, CIFAR-100. We compare SGD, SAM and SGD-TRACER using none of the standard regularizations (no data augmentation, no weight-decay) and a standard training protocol (200 epochs, initial learning rate set to 0.1, cosine learning-rate decay). Furthermore, we randomly flip 50% of the labels so that 50% of the examples are incorrectly labeled.

The noisy label case is of primary interest and relevance in our setting, and relates to both SAM and G-TRACER type schemes which provide robustness w.r.t. weight perturbation since noise can be transferred to the weights. To see this, consider the single-layer case with weight matrix $\boldsymbol{W}$ and input $\boldsymbol{x}$, where we have $L(\boldsymbol{W} + \Delta\boldsymbol{W}, \boldsymbol{x}) = L(\boldsymbol{W}, \boldsymbol{x} + \Delta\boldsymbol{x})$ for the choice $\Delta\boldsymbol{W} = \frac{\boldsymbol{W}\Delta\boldsymbol{x}}{||\boldsymbol{x}||_2^2}\boldsymbol{x}^T$ (indeed, there are infinitely many solutions to the underlying matrix equation), so that input noise robustness corresponds to robustness in weight-space. This type of construction can be generalized to deeper architectures (Seong et al., 2018). Methods like G-TRACER and SAM which are robust to weight perturbations are for this reason expected to be robust to noise. Indeed, the most convincing and striking results in the original SAM paper concern robustness to label noise.

### 4.1.1 CIFAR-100 baseline with augmentation, consistency check

As a baseline and to establish consistency with other results in the literature and in order to demonstrate empirically that our training procedure is such that our models are well-trained, we apply the basic standard data-augmentations (rescaling, random cropping and flipping) together with a ResNet-20 architecture to the CIFAR-100 benchmark.

Table 1: CIFAR-100: ResNet20, accuracy (standard error)

|  | with aug |
| --- | --- |
| SGD | 70.02% (0.36) |
| SAM | 70.33% (0.22) |
| SGD-TRACER | **70.71%** (0.36) |

The exact experimental setting for the vision tasks (unless otherwise indicated) follows standard practice and is as follows: SGD with weight decay/ridge penalty $5 \times 10^{-4}$, momentum 0.9, initial learning rate 0.1, 200 episodes, cosine learning rate decay to 0, batch size 128, global clip-norm= 1.0. The search spaces for the SAM and G-TRACER penalties $\rho$ were chosen by first running on a logarithmic grid of 10 values $[1 \times 10^{-5}, \dots, 1 \times 10^4]$ and then refining the range based on in-sample convergence, in order to span the space of plausible regularization strengths. These results are in line with (in fact, competitive with) the results in (Möllenhoff & Khan, 2022) (Kwon et al., 2021). [22]

### 4.1.2 CIFAR-100 50% noise, no regularization

Having established the baseline, we now consider the challenging setting where we randomly flip 50% of the labels and drop all augmentations (we simply rescale the inputs), and use no weight decay. The results in Table 2 show that GTRACER significantly improves upon SAM in this challenging setting. In Figure 1 we highlight the results for the same problem over different values of the regularization parameter $\rho$. In Figure 2 we compare the training curves on this problem.

Table 2: CIFAR 100: ResNet20, no weight-decay, 50% noise, accuracy (standard error)

|  | no aug |
| --- | --- |
| SGD | 17.5% (2.41) |
| SAM | 34.63% (1.85) |
| SGD-TRACER | **47.55%** (1.51) |

---

[22]As a further consistency check with practice, follow the training protocol (stepwise learning rate decay over 200 episodes, with learning rates $[.1, .02, .004, .0008]$ at $[0, 60, 120, 160]$) in `https://github.com/weiaicunzai/pytorch-cifar100/tree/master?tab=readme-ov-file` with larger architectures, eg ResNet-18 (11M parameters), and match the expected results for SGD, and see similar improvements vs SGD (75.8% accuracy vs 75.1% accuracy) and SAM (75.3% accuracy, $\rho = .05$).

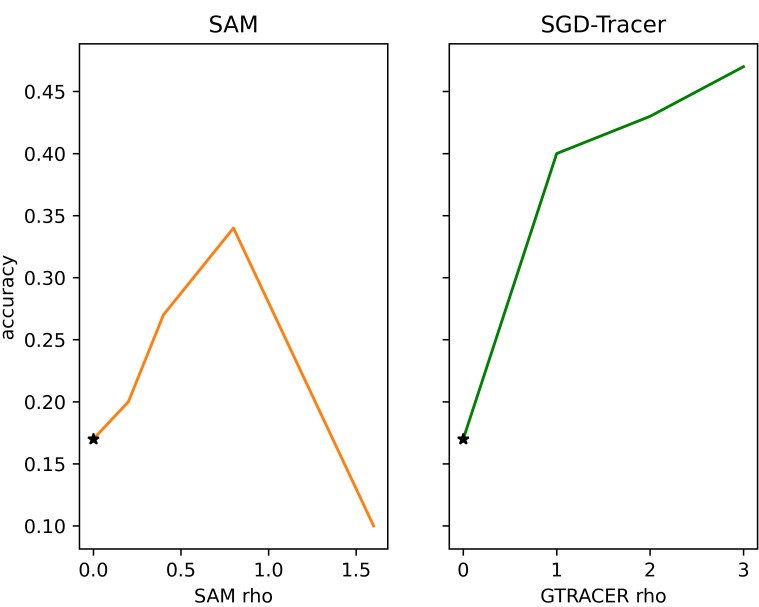

Figure 1: CIFAR 100: ResNet20, no weight-decay, 50% noise, accuracy vs regularization strength. GTRACER dominates the baseline and SAM across a wide range of regularization strengths.

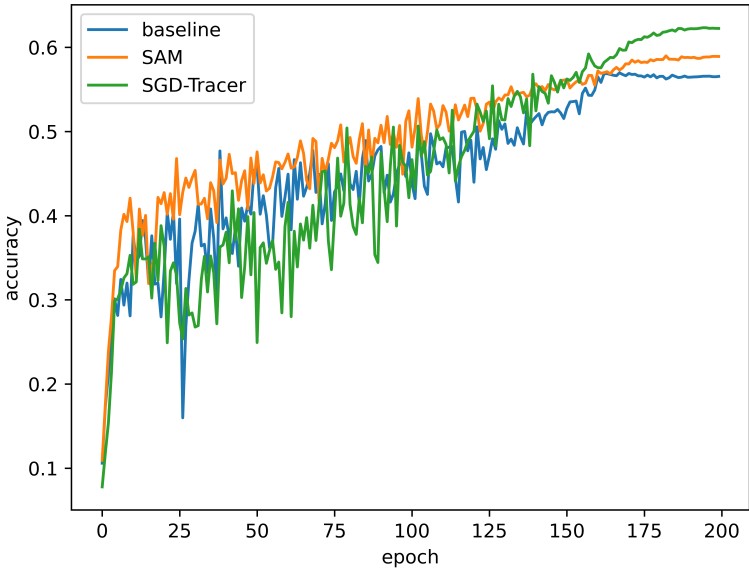

Figure 2: CIFAR 100: ResNet20, 50% noise, test-accuracy training curves. On a standard 200 epoch training protocol with cosine learning-rate decay, SGD-TRACER converges to a solution that generalizes materially better than SGD and SAM

### 4.1.3 CIFAR-100 results with weight decay

We next run SGD-TRACER on CIFAR-100 with and without label noise, with and without augmentation, with random label flipping and with a standard ridge penalty of $5 \times 10^{-4}$. The results in Table 3 show

that SGD-TRACER performs consistently well, with a particularly strong advantage in the presence of noise and/or without additional regularization in the form of data augmentation.

Table 3: CIFAR-100: ResNet20, accuracy (standard error)

|  | **no aug** | **with aug** | **50% noise & no aug** |
|---|---|---|---|
| SGD | 51.43 % (0.41) | 70.02% (0.36) | 21.96% (0.36) |
| SAM | 58.98 % (0.52) | 70.33% (0.22) | 49.89% (0.32) |
| SGD-TRACER | **63.47%** (0.32) | **70.71%** (0.36) | **51.62%** (0.18) |

### 4.1.4 ViT, no pretaining

We now turn to transformer architectures and use the Keras ViTTiny16 vision transformer architecture from the KerasCV library (Wood et al., 2022). We apply the standard augmentations as above with initial learning rate $1 \times 10^{-4}$, and batch size 256. We use this task to investigate the potential for further boosting the performance of G-TRACER by mitigating the gradient magnitude approximation, by splitting each batch into 4 sub-batches, computing squared gradients on each sub-batch and aggregating. [23] For fairness we also compute the SAM gradient on 4 sub-batches and average (as explored in Foret et al. (2020)). We see that this batch splitting delivers strong results for G-TRACER and suggests that moving to per-example gradients could significantly strengthen empirical results. We see that, despite the considerable challenge in

Table 4: CIFAR-100: ViT, accuracy (standard error)

|  | **with aug** |
|---|---|
| SGD | 37.7 % (0.71) |
| SAM | 38.2 % (0.52) |
| SAM batch-split | 38.7 % (0.44) |
| SGD-TRACER | 39.1 % (0.32) |
| SGD-TRACER batch-split | **41.6** % (0.28) |

losing the inductive bias of locality which drives the success of CNNs on vision tasks, SGD-TRACER is able to deliver a 10% performance boost to the naive SGD solution.

## 4.2 NLP

For NLP tasks we use the Huggingface Bert-base-uncased (Devlin et al., 2018) checkpoint together with Adam-TRACER with max sequence length 256 and batch size 8. We fine-tune using Adam-TRACER, using a standard protocol of 5 epochs with initial learning rate $2 \times 10^{-5}$ and decay the learning rate using a linear schedule, with final learning rate $1 \times 10^{-5}$. The search ranges for the SAM and G-TRACER penalty parameters are chosen as in the vision experiments. Each run is repeated 20 times. We choose 3 distinct fine-tuning tasks[24]:

- BoolQ: boolean question answering (Clark et al., 2019)

- WiC: Words in Context (Pilehvar & Camacho-Collados, 2018)

- RTE: Recognizing Textual Entailment (Wang et al., 2020)

and we see that Adam with a G-TRACER performs competitively, and has the additional property of producing more stable results across runs (as reflected in the standard errors). Note that in this setting, the extensive pre-training (Devlin et al., 2018) combined with fine-tuning acts as a strong regularizer and that the relative performance gains we see from using a G-TRACER are smaller than those we observe in the ViT setting without pretraining.

---

[23]Using tools such as (Dangel et al., 2020) would allow per-example squared gradients to be calculated.
[24]All of which have been included in the challenging SuperGlue benchmark

Table 5: NLP tasks BERT base-uncased results, accuracy (standard error)

|             | **BOOLQ**        | **WIC**          | **RTE**          |
|-------------|------------------|------------------|------------------|
| Adam        | 73.84% (0.14)    | 69.36% (0.08)    | 69.18% (0.33)    |
| SAM         | 73.95% (0.13)    | 69.06% (0.07)    | 69.54% (0.28)    |
| Adam-TRACER | **75.09%** (0.04) | **70.01%** (0.06) | **70.13%** (0.18) |

## 5   Conclusion

Motivated by the notable empirical success of SAM, a prior that flat (in expectation, and in an intrinsic, geometric sense) minima should generalize better than sharp minima, and noting the connections between the generalized Bayes objective and SAM, we have derived a new algorithm that is simple to implement and understand, cheap to evaluate, provably convergent, naturally scale-independent (and approximately coordinate-free) and which shows promising performance on standard benchmarks in vision and NLP, and across transformer and convolutional architectures. Performance is notably strong for challenging low signal-to-noise ratio and large batch problems, and in settings where other regularizations (data augmentations, tailored learning rate schedules, weight decay) are not used. Crucially, the algorithm is straightforwardly derived from an approximate natural gradient optimization of an ELBO-type objective and does not rely on the use of small batch sizes (or "m-sharpness" (Foret et al., 2020)) or other poorly understood (and frequently expensive to compute) heuristics.

## A   Appendix

### A.1   G-TRACER detailed derivation

We begin our exposition with a background on generalized variational posteriors and then derive the G-TRACER regularizer by performing natural gradient variational inference.

#### A.1.1   Generalized variational posterior

Our starting point is similar to that of SAM, but uses a more general objective, which arises in the variational optimization of a generalized posterior distribution, $q$, over the space of probability measures $\mathcal{P}(\Theta)$ on the parameter space $\Theta$ given by (Bissiri et al., 2016):

$$q^*(\boldsymbol{w}) = \arg\min_{q \in \mathcal{P}(\Theta)} \left\{ \mathbb{E}_{q(\boldsymbol{w})}[L(\boldsymbol{w})] + \mathbb{D}_{KL}[q,p] \right\} \tag{52}$$

for which, when $Z = \int_{\Theta} \exp\left\{-\sum_{i=1}^{n} l(\boldsymbol{w}, \boldsymbol{x_i})\right\} \pi(\boldsymbol{\theta}) d\boldsymbol{\theta} < \infty$, the solution is given by the generalized posterior:

$$q^*(\boldsymbol{w}) \propto p(\boldsymbol{w}) \prod_{i=1}^{N} \exp\{-l(\boldsymbol{w}, \boldsymbol{x_i})\} \tag{53}$$

The terms $\exp\{-l(\boldsymbol{w}, \boldsymbol{x_i})\}$ are to be interpreted as quasi-likelihoods, and for the particular choice $l(\boldsymbol{w}, \boldsymbol{x_i}) = -\log p(\boldsymbol{x_i}|\boldsymbol{w})$, we recover the standard Bayesian posterior. As this infinite dimensional optimization is in general intractable, it is usual to assume that the posterior belongs to a parametric family $\mathcal{Q} \subset \mathcal{P}$:

$$q^*(\boldsymbol{w}) = \arg\min_{q \in \mathcal{Q}(\Theta)} \left\{ \mathbb{E}_{q(\boldsymbol{w})}[L(\boldsymbol{w})] + \mathbb{D}_{KL}[q,p] \right\} \tag{54}$$

which, for the choice $l(\boldsymbol{w}, \boldsymbol{x_i}) = -\log p(\boldsymbol{x_i}|\boldsymbol{w})$, is the same objective (up to a constant factor) as the evidence lower bound (ELBO) used in variational Bayes.

In practice, it is often found that tempering the KL divergence term by a positive factor $\rho < 1$ produces optimal performance, giving rise to:

$$q^*(\boldsymbol{w}) = \arg\min_{q \in \mathcal{Q}(\Theta)} \left\{ \mathbb{E}_{q(\boldsymbol{w})}[L(\boldsymbol{w})] + \rho \mathbb{D}_{KL}[q,p] \right\} \tag{55}$$

### A.1.2 Derivation of the TRACER flatness-inducing regularizer

Following Khan & Rue (2021) and Zhang et al. (2017), we make the assumption $q(\boldsymbol{w}) \sim \mathcal{N}(\boldsymbol{\mu}, \boldsymbol{\Sigma})$ and seek to optimize the variational objective in equation 55 w.r.t. the variational parameters $\boldsymbol{\phi} = (\boldsymbol{\mu}, \boldsymbol{\Sigma})$ [25] using natural gradient descent. This allows us to derive an algorithm that respects the intrinsic geometry of the parameter space, and thus derive an algorithm that seeks sharp minima in an approximately coordinate-independent way.

Thus our objective is to minimize:

$$L(\boldsymbol{\phi}) := \mathbb{E}_{q(\boldsymbol{w})}[L(\boldsymbol{w})] + \rho \mathbb{D}_{KL}[q, p] \tag{56}$$

w.r.t. $\boldsymbol{\phi}$ where $\rho$ is a positive real-valued regularization parameter.

**Natural gradient overview**

We first introduce and motivate the natural gradient. The negative gradient corresponds to the steepest descent direction in the Euclidean metric:

$$\frac{-\boldsymbol{\nabla}_{\boldsymbol{\phi}} L}{\|\boldsymbol{\nabla}_{\boldsymbol{\phi}} L\|} = \lim_{\epsilon \to 0} \frac{1}{\epsilon} \underset{\boldsymbol{\Delta\phi}: \|\boldsymbol{\Delta\phi}\|_2 < \epsilon}{\operatorname{argmin}} L(\boldsymbol{\phi} + \boldsymbol{\Delta\phi}) \tag{57}$$

and thus depends on the chosen coordinates $\boldsymbol{\phi}$. In contrast, the so-called natural gradient update corresponds to steepest descent in the KL-divergence metric:

$$\frac{-F^{-1} \boldsymbol{\nabla}_{\boldsymbol{\phi}} L}{\|\boldsymbol{\nabla}_{\boldsymbol{\phi}} L\|} = \lim_{\epsilon \to 0} \frac{1}{\epsilon} \underset{\boldsymbol{\Delta\phi}: \mathbb{D}_{KL}[q_{\boldsymbol{\phi}}, q_{\boldsymbol{\phi} + \boldsymbol{\Delta\phi}}] < \epsilon}{\operatorname{argmin}} L(\boldsymbol{\phi} + \boldsymbol{\Delta\phi}) \tag{58}$$

where $F$ is the FIM:

$$F := \mathbb{E}_{q_{\boldsymbol{\phi}}(\boldsymbol{w})} \left[ \boldsymbol{\nabla}_{\boldsymbol{\phi}} \log q_{\boldsymbol{\phi}}(\boldsymbol{w})^T \boldsymbol{\nabla}_{\boldsymbol{\phi}} \log q_{\boldsymbol{\phi}}(\boldsymbol{w}) \right] = \mathbb{E}_{q_{\boldsymbol{\phi}}(\boldsymbol{w})} \left[ -\boldsymbol{\nabla}_{\boldsymbol{\phi}}^2 \log q_{\boldsymbol{\phi}}(\boldsymbol{w}) \right] \tag{59}$$

which defines a Riemannian metric on the variational parameter manifold. Expanding to second order in a small neighborhood of $\boldsymbol{\phi}$ we have:

$$\mathbb{D}_{KL}[q_{\boldsymbol{\phi}}, q_{\boldsymbol{\phi} + \boldsymbol{\Delta\phi}}] = \mathbb{E}_{q_{\boldsymbol{\phi}}(\boldsymbol{w})} \left[ -\boldsymbol{\Delta\phi}^T \boldsymbol{\nabla}_{\boldsymbol{\phi}} \log q_{\boldsymbol{\phi}}(\boldsymbol{w}) - \frac{1}{2} \boldsymbol{\Delta\phi}^T \boldsymbol{\nabla}_{\boldsymbol{\phi}}^2 \log q_{\boldsymbol{\phi}}(\boldsymbol{w}) \boldsymbol{\Delta\phi} \right] + O(\|\boldsymbol{\Delta\phi}\|^3) \tag{60}$$

and since:

$$\mathbb{E}_{q_{\boldsymbol{\phi}}(\boldsymbol{w})} \boldsymbol{\nabla}_{\boldsymbol{\phi}} \log q_{\boldsymbol{\phi}}(\boldsymbol{w}) = \mathbb{E}_{q_{\boldsymbol{\phi}}(\boldsymbol{w})} \left[ \frac{\boldsymbol{\nabla}_{\boldsymbol{\phi}} q_{\boldsymbol{\phi}}(\boldsymbol{w})}{q_{\boldsymbol{\phi}}(\boldsymbol{w})} \right] = \boldsymbol{\nabla}_{\boldsymbol{\phi}} \mathbb{E}_{q_{\boldsymbol{\phi}}(\boldsymbol{w})}[1] = 0 \tag{61}$$

the FIM (under certain regularity conditions) can be seen to be the Hessian (or curvature) of the K-L divergence:

$$\mathbb{D}_{KL}[q_{\boldsymbol{\phi}}, q_{\boldsymbol{\phi} + \boldsymbol{\Delta\phi}}] = -\frac{1}{2} \boldsymbol{\Delta\phi}^T \mathbb{E}_{q_{\boldsymbol{\phi}}(\boldsymbol{w})} \left[ \boldsymbol{\nabla}_{\boldsymbol{\phi}}^2 \log q_{\boldsymbol{\phi}}(\boldsymbol{w}) \right] \boldsymbol{\Delta\phi} + O(\|\boldsymbol{\Delta\phi}\|^3) = \frac{1}{2} \boldsymbol{\Delta\phi}^T F \boldsymbol{\Delta\phi} + O(\|\boldsymbol{\Delta\phi}\|^3) \tag{62}$$

**Natural gradient update**

It turns out that the update equations have a particularly simple form when $q_{\boldsymbol{\phi}}(\boldsymbol{w})$ is parameterized as $\boldsymbol{\phi} = (\boldsymbol{\mu}, \boldsymbol{\Lambda}^{-1})$, and the following proposition gives expressions for the natural gradient vectors of our objective (56) w.r.t. the mean and precision (for proof see Appendix A.5):

**Proposition 1.** *For a probability distribution with pdf $q_{\boldsymbol{\phi}}(\boldsymbol{w}) \sim \mathcal{N}(\boldsymbol{\mu}, \boldsymbol{\Lambda}^{-1})$ with the parameterization $\boldsymbol{\phi} = (\boldsymbol{\mu}, \boldsymbol{\Lambda}^{-1})$, the natural gradients of $L$ w.r.t. $\boldsymbol{\mu}$ and $\boldsymbol{\Lambda}$ of are given by:*

$$\begin{aligned} \tilde{\boldsymbol{\nabla}}_{\boldsymbol{\mu}} L &= \boldsymbol{\Sigma} \mathbb{E}_q[\boldsymbol{\nabla}_{\boldsymbol{w}} L(\boldsymbol{w}) + \rho \boldsymbol{\nabla}_{\boldsymbol{w}} p(\boldsymbol{w})] \\ \tilde{\boldsymbol{\nabla}}_{\boldsymbol{\Lambda}} L &= -\mathbb{E}_q[\boldsymbol{\nabla}_{\boldsymbol{w}}^2 L(\boldsymbol{w}) - \rho \boldsymbol{\nabla}_{\boldsymbol{w}}^2 p(\boldsymbol{w})] + \rho \boldsymbol{\Sigma}^{-1} \end{aligned} \tag{63}$$

---

[25] We will write, to keep the notation as light as possible, the set of variational parameters as $\boldsymbol{\phi} = \{\boldsymbol{\mu}, \boldsymbol{\Sigma}\}$. Depending on the context, e.g. when we write the gradient $\boldsymbol{\nabla}_{\boldsymbol{\phi}}$, we will take this to mean $\boldsymbol{\phi} = \begin{bmatrix} \boldsymbol{\mu} \\ \text{vec}(\boldsymbol{\Sigma}) \end{bmatrix}$

Assuming an isotropic Gaussian prior, $p(\boldsymbol{w}) \sim \mathcal{N}(\boldsymbol{0}, \eta \boldsymbol{I})$, performing gradient descent w.r.t. this natural gradient then leads to the following iterative update equations:

$$\boldsymbol{\mu} \leftarrow \boldsymbol{\mu} - \alpha \boldsymbol{\Lambda}^{-1} \left( \mathbb{E}_q[\boldsymbol{\nabla_w} L(\boldsymbol{w})] + \frac{\rho}{\eta} \boldsymbol{w} \right)$$
$$\boldsymbol{\Lambda} \leftarrow (1 - \beta)\boldsymbol{\Lambda} + \beta \left( \frac{\mathbb{E}_q[\boldsymbol{\nabla_w^2} L(\boldsymbol{w})]}{\rho} + \eta^{-1} \boldsymbol{I} \right)$$
(64)

where $\alpha$ and $\beta$ are the learning rates for the mean and precision updates, respectively. We work with each of these update equations in turn. Starting with the update equation for the mean $\boldsymbol{\mu}$, the key observation is that the expectation $\mathbb{E}_q[\boldsymbol{\nabla_w} L(\boldsymbol{w})]$ is taken with respect to the distribution $q(\boldsymbol{w})$, which is an exponential moving average of the expected Hessian $\mathbb{E}_q[\boldsymbol{\nabla_w^2} L(\boldsymbol{w})]$. This updating happens naturally as a consequence of taking natural gradient steps, and leads to an approximately coordinate-free algorithm in the sequel. Applying Bonnet's theorem (Khan & Rue, 2021) and forming the second-order approximation to the loss we obtain:

$$\mathbb{E}_q[\boldsymbol{\nabla_w} L(\boldsymbol{w})] = \boldsymbol{\nabla_\mu} \mathbb{E}_q[L(\boldsymbol{w})] \approx \boldsymbol{\nabla_\mu} \mathbb{E}_q[L(\boldsymbol{\mu}) + \frac{1}{2}(\boldsymbol{w} - \boldsymbol{\mu})^T \boldsymbol{\nabla_w^2} L(\boldsymbol{w})|_{\boldsymbol{w}=\boldsymbol{\mu}}(\boldsymbol{w} - \boldsymbol{\mu})]$$
(65)

We also have:

$$\mathbb{E}_q[(\boldsymbol{w} - \boldsymbol{\mu})^T \boldsymbol{\nabla_w^2} L(\boldsymbol{w})|_{\boldsymbol{w}=\boldsymbol{\mu}}(\boldsymbol{w} - \boldsymbol{\mu})] = \mathbb{E}_q[\text{Tr}\left((\boldsymbol{w} - \boldsymbol{\mu})^T \boldsymbol{\nabla_w^2} L(\boldsymbol{w})|_{\boldsymbol{w}=\boldsymbol{\mu}}(\boldsymbol{w} - \boldsymbol{\mu})\right)] = \text{Tr}(\boldsymbol{\Sigma H})$$
(66)

where $\boldsymbol{H}$ is the Hessian $\boldsymbol{\nabla_w^2} L(\boldsymbol{w})$. We therefore have that:

$$\mathbb{E}_q[\boldsymbol{\nabla_w} L(\boldsymbol{w})] \approx \boldsymbol{\nabla_\mu}[L(\boldsymbol{\mu}) + \frac{1}{2}\text{Tr}(\boldsymbol{\Sigma H})]$$
(67)

Choosing the prior variance $\eta$ to be infinite and thus ignoring terms involving $\eta$ in both update equations (corresponding to an improper prior, and so consistent with the discussion above), leads to the following update for the mean:

$$\boldsymbol{\mu} \leftarrow \boldsymbol{\mu} + \alpha \boldsymbol{\Lambda} \left( \boldsymbol{\nabla_\mu}[L(\boldsymbol{\mu}) + \frac{1}{2}\text{Tr}(\boldsymbol{\Sigma H})] \right)$$
(68)

Thus, in order to blur the loss with multivariate Gaussian noise in a way that aligns with the intrinsic geometry of the parameter space, we can (to second order) augment the loss with a term involving the trace of the Hessian. Considering now the update equation for the precision, we can use Price's theorem (Khan & Rue, 2021) together with a Taylor expansion to get, to second order $\mathbb{E}_q[\boldsymbol{\nabla_w^2} L(\boldsymbol{w})] \approx \boldsymbol{\nabla_w^2} L(\boldsymbol{w})|_{\boldsymbol{w}=\boldsymbol{\mu}}$ (see Appendix A.3 for details), which leads to

$$\boldsymbol{\Lambda} \leftarrow (1 - \beta)\boldsymbol{\Lambda} + \beta \left( \frac{\boldsymbol{\nabla_w^2} L(\boldsymbol{w})|_{\boldsymbol{w}=\boldsymbol{\mu}}}{\rho} \right)$$
(69)

We next substitute, as is common in the literature using approximate second order approximation (Martens, 2020), the log-likelihood Fisher Information Matrix $\boldsymbol{F}$ for the Hessian in the update equation for the precision, where:

$$\boldsymbol{F} = \nabla_{\boldsymbol{w}'}^2 \mathbb{D}_{KL}[p(\boldsymbol{y}|\boldsymbol{x}, \boldsymbol{w}), p(\boldsymbol{y}|\boldsymbol{x}, \boldsymbol{w}')]|_{\boldsymbol{w}'=\boldsymbol{w}}$$
$$= \mathbb{E}_{\boldsymbol{y} \sim p(\boldsymbol{y}|\boldsymbol{x}, \boldsymbol{w}), \boldsymbol{x} \sim p(\boldsymbol{x})}[(\nabla \log p(\boldsymbol{y}|\boldsymbol{x}, \boldsymbol{w}))(\nabla \log p(\boldsymbol{y}|\boldsymbol{x}, \boldsymbol{w}))^T]$$
(70)

We pass to the underlying parameter space by taking the maximum a posteriori (MAP) estimate, write the update in terms of $\boldsymbol{G} := \rho \boldsymbol{\Lambda}$, absorbing constants into $\rho$ and $\alpha$, and write the iteration in terms of the parameter $\boldsymbol{w}$, and thus obtain the general update equations 18:

$$\boldsymbol{w} \leftarrow \boldsymbol{w} - \alpha \left( \boldsymbol{\nabla_w}[L(\boldsymbol{w}) + \rho\text{Tr}(\boldsymbol{G}^{-1}\boldsymbol{H})] \right)$$
$$\boldsymbol{G} \leftarrow (1 - \beta)\boldsymbol{G} + \beta \boldsymbol{F}$$
(71)

Crucially, the penalty term $\rho\text{Tr}(\boldsymbol{G}^{-1}\boldsymbol{H})$ can be seen to be invariant to affine coordinate transformations. Indeed, under an affine coordinate transformation with Jacobian $\boldsymbol{J}$, we have $\boldsymbol{H} \to \boldsymbol{J}^T \boldsymbol{H}' \boldsymbol{J}$ and $\boldsymbol{G} \to \boldsymbol{J}^T \boldsymbol{G}' \boldsymbol{J}$ so that:

$$\text{Tr}\left(\boldsymbol{G}^{-1}\boldsymbol{H}\right) = \text{Tr}\left(\boldsymbol{J}^{-1}\boldsymbol{G}'^{-1}\boldsymbol{J}^{T^{-1}}\boldsymbol{J}^T \boldsymbol{H}' \boldsymbol{J}\right) = \text{Tr}\left(\boldsymbol{J}^{-1}\boldsymbol{G}'^{-1}\boldsymbol{H}'\boldsymbol{J}\right) = \text{Tr}\left(\boldsymbol{G}'^{-1}\boldsymbol{H}'\right)$$
(72)

More generally, given a smooth coordinate change defined by a diffemorphism $\Phi : \mathbb{R}^p \to \mathbb{R}^p$ and Jacobian $J(\boldsymbol{w})$, then, given sufficiently rapid exponential decay in the update equation for the Fisher, subject to $\Phi$ having sufficient regularity, the penalty term is readily seen to be approximately coordinate free. Crucially, in the neighborhood of a minimum, where the Hessian and G transform as (0,2) tensors we also have invariance to general coordinate transformations (Kristiadi et al., 2023).

Although the evaluation of the GGN matrix, in particular the matrix multiplications involving the Jacobians $J_f$, can be relatively costly, the FIM can be expressed as an expectation of outer products of gradients w.r.t. the output distribution $p(\boldsymbol{y}|\boldsymbol{x},\boldsymbol{w})$:

$$\frac{1}{n}\sum_{i=1}^{n}\mathbb{E}_{p(\boldsymbol{y}|\boldsymbol{x_i},\boldsymbol{w})}\left[\boldsymbol{\nabla_w}\log p(\boldsymbol{y}|\boldsymbol{x_i},\boldsymbol{w})^T\boldsymbol{\nabla_w}\log p(\boldsymbol{y}|\boldsymbol{x_i},\boldsymbol{w})\right] \approx \frac{1}{n}\sum_{i=1}^{n}\boldsymbol{\nabla_w}\log p(\tilde{\boldsymbol{x}_i}|\boldsymbol{x_i},\boldsymbol{w})^T\boldsymbol{\nabla_w}\log p(\tilde{\boldsymbol{y}_i}|\boldsymbol{x_i},\boldsymbol{w})$$

$$(73)$$

which, following Martens (2020), can be estimated using a single Monte Carlo sample from the output distribution: $\tilde{\boldsymbol{y}} \sim p(\boldsymbol{y}|\boldsymbol{x_i},\boldsymbol{w})$. Using this (biased) Fisher approximation in our setting thus requires gradients to be calculated through an expectation $\boldsymbol{\nabla_w}\mathbb{E}_{p(\boldsymbol{y}|\boldsymbol{x},\boldsymbol{w})}[L(\boldsymbol{w};\boldsymbol{y})]$, approximated using a Monte Carlo sample from the model's output distribution. Since the expectation is taken w.r.t. a distribution which depends on $\boldsymbol{w}$, it is necessary to reparameterize so that the discrete Monte Carlo sample is expressed as the deterministic transformation of a $g_{\boldsymbol{w}}(\boldsymbol{z})$ (depending on $\boldsymbol{w}$) of a sample $\boldsymbol{z} \sim h_{\boldsymbol{\theta}}(\boldsymbol{z})$ from a distribution not depending on $\boldsymbol{w}$, so that $\mathbb{E}_{p(\boldsymbol{y}|\boldsymbol{x},\boldsymbol{w})}[L(\boldsymbol{w};\boldsymbol{y})] = \mathbb{E}_{\boldsymbol{z}\sim h_{\boldsymbol{\theta}}(\boldsymbol{z})}[L(\boldsymbol{w};g_{\boldsymbol{w}}(\boldsymbol{z}))]$. In the discrete case (corresponding to classification), since the argmax function is non-differentiable, the standard approach is the Gumbel-Softmax reparameterization (Jang et al., 2016), which uses the softmax function as a continuous relaxation of the argmax function together with i.i.d. samples distributed as Gumbel(0,1).

It is important to note that this approach is different from simply evaluating $\log p(\boldsymbol{y}|\boldsymbol{x},\boldsymbol{w})$ on the training labels, a widely used approximation known as the empirical Fisher $\tilde{\boldsymbol{F}}$:

$$\tilde{\boldsymbol{F}} := \sum_{i=1}^{n}\boldsymbol{\nabla_w}\log p(\boldsymbol{y_i}|\boldsymbol{x_i},\boldsymbol{w})^T\boldsymbol{\nabla_w}\log p(\boldsymbol{y_i}|\boldsymbol{x_i},\boldsymbol{w}) \qquad (74)$$

This, despite lacking the same convergence guarantees, performs competitively in many settings (Kunstner et al., 2019). We find in our experiments that the empirical Fisher performs competitively with the MC approximation to the GGN (Khan et al., 2018; Kingma & Ba, 2014) and has the advantage of being straightforward and cheap to compute from already computed gradients (in the case of Adam-TRACER, the smooth squared gradients are already computed and maintained for use as a preconditioner). Given the conceptual and computational simplicity of this approach we substitute the empirical Fisher for the Fisher Information Matrix, as well as for the Hessian in the update equation for the mean, giving the update equations:

$$\boldsymbol{w} \leftarrow \boldsymbol{w} - \alpha\boldsymbol{\nabla_w}[L(\boldsymbol{w}) + \rho\text{Tr}(\boldsymbol{G^{-1}}\tilde{\boldsymbol{F}})]$$
$$\boldsymbol{G} \leftarrow (1-\beta)\boldsymbol{G} + \beta\tilde{\boldsymbol{F}} \qquad (75)$$

Recent advances in approximate second-order methods in optimization, notably Yao et al. (2020), suggest avenues for improvement, and we leave investigations of alternatives, such as the smoothed (Hessian-free) Hessian diagonal sketch used in AdaHessian, for future work.

We now make two simplifications. First, we use a mean-field approximation, representing the FIM by its diagonal, as is done in Adam (Kingma & Ba, 2014), Adagrad (Duchi et al., 2011) and RMSProp thus:

$$\boldsymbol{F_{\text{diag}}} \approx \frac{1}{n}\sum_{i=1}^{n}\boldsymbol{\nabla_w}\log p(\boldsymbol{y_i}|\boldsymbol{x_i},\boldsymbol{w})^2 \qquad (76)$$

Secondly, it is standard practice to (Martens, 2020) to add Tikhonov regularization or damping via a small positive real constant $\delta$ when using 2nd-order optimization methods, giving in this case the preconditioner: $(\boldsymbol{F_e} + \delta\boldsymbol{I})^{-1}$. In fact this would arise naturally in our setup by choosing $\eta$ to be non-zero, in which case we would simply have $\delta := \frac{\rho}{\eta}$. From an optimization perspective, it is justified by recognizing that the local quadratic model from which the second-order update is ultimately derived is a second-order approximation

to the KL divergence and is thus only valid locally. For directions corresponding to small eigenvalues, parameter updates can lie outside the region where the approximation is reasonable (Martens, 2020). This is true, a fortiori, when diagonal approximations are used, as is the case here. As our emphasis here is on geometric regularization, we drop the preconditioner entirely by choosing $\delta$ to be sufficiently large that the preconditioner is equal to the identity (up to a constant, which is absorbed into the learning rate).

Finally, as most current deep learning frameworks do not straightforwardly support access to per-example gradients, which can in principle be achieved with negligible additional cost (see, for example, BackPACK Dangel et al. (2020) second-order Pytorch extensions), for simplicity and efficiency, we use the gradient magnitude (GM) approximation (Bottou et al., 2016), as used in standard optimizers such as Adam, Adagrad, and RMSprop, replacing the sum of squared gradients with the square of summed gradients:

$$\frac{1}{n} \sum_{i=1}^{n} [\boldsymbol{\nabla}_{\boldsymbol{w}} \log p(\boldsymbol{y_i}|\boldsymbol{x_i}, \boldsymbol{w})]^2 \approx \left[ \frac{1}{n} \sum_{i=1}^{n} \boldsymbol{\nabla}_{\boldsymbol{w}} \log p(\boldsymbol{y_i}|\boldsymbol{x_i}, \boldsymbol{w}) \right]^2 \tag{77}$$

The theoretical justification for this is given in Theorem 1 of (Khan et al., 2018) who show that the sum of squared gradients is close to the squared sum of gradients if the batch and population estimates are sufficiently close.

Writing the resulting FIM diagonal as $(\boldsymbol{\nabla}_{\boldsymbol{w}} L(\boldsymbol{w}))^2$, and using stochastic gradient updates computed on on minibatches $\mathcal{B}$ of the data, $\boldsymbol{\nabla}_{\boldsymbol{w}} L_{\mathcal{B}}(\boldsymbol{w})$, we finally end up with the following update (Algorithm 1):

$$\boldsymbol{w}_{t+1} = \boldsymbol{w}_t - \alpha_t \boldsymbol{\nabla}_{\boldsymbol{w}} \left[ L_{\mathcal{B}}(\boldsymbol{w}_t) + \rho \left( \boldsymbol{\nabla}_{\boldsymbol{w}} L_{\mathcal{B}}(\boldsymbol{w}_t) \right)^2 \cdot 1/(\boldsymbol{f}_t + \delta) \right]$$
$$\boldsymbol{f}_{t+1} = (1 - \beta) \boldsymbol{f}_t + \beta \left( \boldsymbol{\nabla}_{\boldsymbol{w}} L_{\mathcal{B}}(\boldsymbol{w}_t) \right)^2 \tag{78}$$

We show in Appendix A.6 that the algorithm converges to a neighborhood of a local minimum of $L(\boldsymbol{w})$ of size $\mathcal{O}(\rho^2)$. We note in passing that, in this simplest form (after applying the gradient magnitude approximation), the update equations amount to regularizing with a (scale-adjusted) gradient norm. In principle (particularly for the large batch case) we would expect to see significant improvements by moving to per-gradient calculations (which are theoretically no more expensive to compute but require additional work under most current autodiff frameworks).

## A.2   Multivariate Gaussian Fisher Information Matrix

For a multivariate Gaussian with pdf:

$$q(\boldsymbol{x}; \boldsymbol{\mu}, \boldsymbol{\Sigma}) = \frac{1}{(2\pi)^{d/2} |\boldsymbol{\Sigma}|^{1/2}} \exp\left( -\frac{1}{2} (\boldsymbol{x} - \boldsymbol{\mu})^T \boldsymbol{\Sigma}^{-1} (\boldsymbol{x} - \boldsymbol{\mu}) \right)$$

the Fisher Information Matrix with respect to the mean vector $\boldsymbol{\mu}$ is given by:

$$I(\boldsymbol{\mu}) = \mathbb{E}\left[ \left( \frac{\partial \log q(\boldsymbol{x}; \boldsymbol{\mu}, \boldsymbol{\Sigma})}{\partial \boldsymbol{\mu}} \right) \left( \frac{\partial \log q(\boldsymbol{x}; \boldsymbol{\mu}, \boldsymbol{\Sigma})}{\partial \boldsymbol{\mu}} \right)^T \right]$$

First, we compute the partial derivative of the log-likelihood with respect to $\boldsymbol{\mu}$:

$$\frac{\partial \log q(\boldsymbol{x}; \boldsymbol{\mu}, \boldsymbol{\Sigma})}{\partial \boldsymbol{\mu}} = \boldsymbol{\Sigma}^{-1} (\boldsymbol{x} - \boldsymbol{\mu})$$

The Fisher Information Matrix with respect to $\boldsymbol{\mu}$ is then:

$$I(\boldsymbol{\mu}) = \mathbb{E}\left[ \boldsymbol{\Sigma}^{-1} (\boldsymbol{x} - \boldsymbol{\mu})(\boldsymbol{x} - \boldsymbol{\mu})^T \boldsymbol{\Sigma}^{-1} \right]$$

Given that $\boldsymbol{x} \sim \mathcal{N}(\boldsymbol{\mu}, \boldsymbol{\Sigma})$, the expected value $\mathbb{E}[(\boldsymbol{x} - \boldsymbol{\mu})(\boldsymbol{x} - \boldsymbol{\mu})^T] = \boldsymbol{\Sigma}$. Therefore:

$$I(\boldsymbol{\mu}) = \boldsymbol{\Sigma}^{-1} \boldsymbol{\Sigma} \boldsymbol{\Sigma}^{-1} = \boldsymbol{\Sigma}^{-1}$$

The Fisher Information Matrix with respect to the covariance matrix $\boldsymbol{\Sigma}$ is more complex and involves the derivatives of the log-likelihood function with respect to $\boldsymbol{\Sigma}$. The partial derivative of the log-likelihood with respect to $\boldsymbol{\Sigma}$ is:

$$\frac{\partial \log q(\boldsymbol{x}; \boldsymbol{\mu}, \boldsymbol{\Sigma})}{\partial \boldsymbol{\Sigma}} = \frac{1}{2} \left( \boldsymbol{\Sigma}^{-1} (\boldsymbol{x} - \boldsymbol{\mu})(\boldsymbol{x} - \boldsymbol{\mu})^T \boldsymbol{\Sigma}^{-1} - \boldsymbol{\Sigma}^{-1} \right)$$

The Fisher Information Matrix with respect to $\boldsymbol{\Sigma}$ is given by:

$$I(\boldsymbol{\Sigma}) = \mathbb{E}\left[ \left( \frac{\partial \log q(\boldsymbol{x}; \boldsymbol{\mu}, \boldsymbol{\Sigma})}{\partial \boldsymbol{\Sigma}} \right) \left( \frac{\partial \log q(\boldsymbol{x}; \boldsymbol{\mu}, \boldsymbol{\Sigma})}{\partial \boldsymbol{\Sigma}} \right)^T \right]$$

which can be written as:

$$I(\boldsymbol{\Sigma})_{ijkl} = \frac{1}{2} (\boldsymbol{\Sigma}^{-1})_{ik} (\boldsymbol{\Sigma}^{-1})_{jl} + \frac{1}{2} (\boldsymbol{\Sigma}^{-1})_{il} (\boldsymbol{\Sigma}^{-1})_{jk}$$

This can also be written using the Kronecker product and vectorization as:

$$I(\boldsymbol{\Sigma}) = \frac{1}{2} (\boldsymbol{\Sigma}^{-1} \otimes \boldsymbol{\Sigma}^{-1})$$

**Combined Fisher Information Matrix**

For both $\boldsymbol{\mu}$ and $\boldsymbol{\Sigma}$, the combined Fisher Information Matrix can be represented as a block matrix:

$$I(\boldsymbol{\phi}) = \begin{bmatrix} I(\boldsymbol{\mu}) & 0 \\ 0 & I(\boldsymbol{\Sigma}) \end{bmatrix}$$

where $\boldsymbol{\phi} = (\boldsymbol{\mu}, \mathrm{vec}(\boldsymbol{\Sigma}))$.

In summary, the Fisher Information Matrix for a multivariate Gaussian with mean $\boldsymbol{\mu}$ and covariance $\boldsymbol{\Sigma}$ is:

$$I(\boldsymbol{\theta}) = \begin{bmatrix} \boldsymbol{\Sigma}^{-1} & 0 \\ 0 & \frac{1}{2}(\boldsymbol{\Sigma}^{-1} \otimes \boldsymbol{\Sigma}^{-1}) \end{bmatrix}$$

### A.3 Approximate expected Hessian

**Lemma 1.** *To second order, we can approximate the expected Hessian w.r.t. a multivariate Gaussian with pdf: $q(x) \sim \mathcal{N}(\boldsymbol{\mu}, \boldsymbol{\Lambda}^{-1})$ by its value at the mean:*

$$\mathbb{E}_q[\boldsymbol{\nabla}_{\boldsymbol{w}}^2 L(\boldsymbol{w})] \approx \boldsymbol{\nabla}_{\boldsymbol{w}}^2 L(\boldsymbol{w})|_{\boldsymbol{w}=\boldsymbol{\mu}} \tag{79}$$

*Proof.* Following Khan & Rue (2021), by Price's theorem: $\boldsymbol{\nabla}_{\boldsymbol{\Sigma}} \mathbb{E}_q[L(\boldsymbol{w})] = \frac{1}{2}\mathbb{E}_q[\boldsymbol{\nabla}_{\boldsymbol{w}}^2 L(\boldsymbol{w})]$, we have:

$$\mathbb{E}_q[\boldsymbol{\nabla}_{\boldsymbol{w}}^2 L(\boldsymbol{w})] = 2\boldsymbol{\nabla}_{\boldsymbol{\Lambda}^{-1}}^2 \mathbb{E}_q[L(\boldsymbol{w})] \tag{80}$$

expanding the r.h.s. to second order using a Taylor series, this is equivalent to:

$$2\boldsymbol{\nabla}_{\boldsymbol{\Lambda}^{-1}}^2 \mathbb{E}_q[(\boldsymbol{w} - \boldsymbol{\mu})^T \boldsymbol{\nabla}_{\boldsymbol{w}}^2 L(\boldsymbol{w})|_{\boldsymbol{w}=\boldsymbol{\mu}} (\boldsymbol{w} - \boldsymbol{\mu})] \tag{81}$$

Finally, noting that $\mathbb{E}_q[(\boldsymbol{w} - \boldsymbol{\mu})^T \boldsymbol{\nabla}_{\boldsymbol{w}}^2 L(\boldsymbol{w})|_{\boldsymbol{w}=\boldsymbol{\mu}} (\boldsymbol{w} - \boldsymbol{\mu})] = \mathrm{Tr}\left[\frac{1}{2}\boldsymbol{\Lambda}^{-1} \boldsymbol{\nabla}_{\boldsymbol{w}}^2 L(\boldsymbol{w})|_{\boldsymbol{w}=\boldsymbol{\mu}}\right]$, we have, to second order:

$$\mathbb{E}_q[\boldsymbol{\nabla}_{\boldsymbol{w}}^2 L(\boldsymbol{w})] \approx 2\boldsymbol{\nabla}_{\boldsymbol{\Lambda}^{-1}}^2 \mathrm{Tr}\left[\frac{1}{2}\boldsymbol{\Lambda}^{-1} \boldsymbol{\nabla}_{\boldsymbol{w}}^2 L(\boldsymbol{w})|_{\boldsymbol{w}=\boldsymbol{\mu}}\right] = \boldsymbol{\nabla}_{\boldsymbol{w}}^2 L(\boldsymbol{w})|_{\boldsymbol{w}=\boldsymbol{\mu}} \tag{82}$$

$\square$

### A.4 Objective function gradient

**Lemma 2.** *The gradient of the objective 56 towards* $\phi' = \begin{bmatrix} \boldsymbol{\mu} \\ \mathrm{vec}(\boldsymbol{\Sigma}) \end{bmatrix}$ *is given by:*

$$\boldsymbol{\nabla}_{\boldsymbol{\mu}}L = \mathbb{E}_q[\boldsymbol{\nabla}_{\boldsymbol{w}}L(\boldsymbol{w}) - \rho\boldsymbol{\nabla}_{\boldsymbol{w}}\log p(\boldsymbol{w})] \tag{83}$$

$$\boldsymbol{\nabla}_{\boldsymbol{\Sigma}}L = \frac{1}{2}\mathbb{E}_q[\boldsymbol{\nabla}_{\boldsymbol{w}}^2 L(\boldsymbol{w}) - \rho\boldsymbol{\nabla}_{\boldsymbol{w}}^2\log p(\boldsymbol{w})] - \frac{\rho}{2}\boldsymbol{\Sigma}^{-1} \tag{84}$$

*Proof.* We will make use of Bonnet's and Price's theorems (Khan & Rue, 2021), which, given an expectation w.r.t. a multivariate Gaussian and twice-differentiability of $L$, allows us to interchange expectations and gradients as follows:

$$\begin{aligned} \boldsymbol{\nabla}_{\boldsymbol{\mu}}\mathbb{E}_q[L(\boldsymbol{w})] &= \mathbb{E}_q[\boldsymbol{\nabla}_{\boldsymbol{w}}L(\boldsymbol{w})] \\ \boldsymbol{\nabla}_{\boldsymbol{\Sigma}}\mathbb{E}_q[L(\boldsymbol{w})] &= \frac{1}{2}\mathbb{E}_q[\boldsymbol{\nabla}_{\boldsymbol{w}}^2 L(\boldsymbol{w})] \end{aligned} \tag{85}$$

Taking the negative gradient of the objective w.r.t. to $\boldsymbol{\mu}$, and applying Bonnet's theorem, and the fact that the expectation of the score is 0, we have:

$$\boldsymbol{\nabla}_{\boldsymbol{\mu}}\left(\mathbb{E}_q[L(\boldsymbol{w})] + \rho\mathbb{D}_{KL}[q(\boldsymbol{w}), p(\boldsymbol{w})]\right) = \mathbb{E}_q[\boldsymbol{\nabla}_{\boldsymbol{w}}L(\boldsymbol{w})] - \rho\mathbb{E}_q\left[\boldsymbol{\nabla}_{\boldsymbol{w}}\log p(\boldsymbol{w})\right] \tag{86}$$

Taking the gradient w.r.t. $\boldsymbol{\Sigma}$, and applying Price's theorem, we have:

$$\boldsymbol{\nabla}_{\boldsymbol{\Sigma}}\left(\mathbb{E}_q[L(\boldsymbol{w})] + \rho\mathbb{D}_{KL}[q(\boldsymbol{w}), p(\boldsymbol{w})]\right) = \frac{1}{2}\mathbb{E}_q\left[\boldsymbol{\nabla}_{\boldsymbol{w}}^2 L(\boldsymbol{w}) + \rho\boldsymbol{\nabla}_{\boldsymbol{w}}^2\log q(\boldsymbol{w}) - \rho\boldsymbol{\nabla}_{\boldsymbol{w}}^2\log p(\boldsymbol{w})\right] \tag{87}$$

and since:

$$\mathbb{E}_q\left[\boldsymbol{\nabla}_{\boldsymbol{w}}^2\log q(\boldsymbol{w})\right] = -\frac{1}{2}\mathbb{E}_q\left[\boldsymbol{\nabla}_{\boldsymbol{w}}^2\left(\log|\boldsymbol{\Sigma}| + (\boldsymbol{w}-\boldsymbol{\mu})^T\boldsymbol{\Sigma}^{-1}(\boldsymbol{w}-\boldsymbol{\mu})\right)\right] = -\boldsymbol{\Sigma}^{-1} \tag{88}$$

We obtain

$$\boldsymbol{\nabla}_{\boldsymbol{\mu}}L = \mathbb{E}_q[\boldsymbol{\nabla}_{\boldsymbol{w}}L(\boldsymbol{w}) - \rho\boldsymbol{\nabla}_{\boldsymbol{w}}\log p(\boldsymbol{w})] \tag{89}$$

$$\boldsymbol{\nabla}_{\boldsymbol{\Sigma}}L = \frac{1}{2}\mathbb{E}_q[\boldsymbol{\nabla}_{\boldsymbol{w}}^2 L(\boldsymbol{w}) - \rho\boldsymbol{\nabla}_{\boldsymbol{w}}^2\log p(\boldsymbol{w})] - \frac{\rho}{2}\boldsymbol{\Sigma}^{-1} \tag{90}$$

$\square$

### A.5 Objective function natural gradient

**Proposition 2.** *The natural gradients of the objective 56 w.r.t. the parameters* $\phi = (\boldsymbol{\mu}, \boldsymbol{\Lambda})$ *of* $q(x) \sim \mathcal{N}(\boldsymbol{\mu}, \boldsymbol{\Lambda}^{-1})$ *are given by:*

$$\tilde{\boldsymbol{\nabla}}_{\boldsymbol{\mu}}L = \boldsymbol{\Sigma}\mathbb{E}_q[\boldsymbol{\nabla}_{\boldsymbol{w}}L(\boldsymbol{w}) + \rho\boldsymbol{\nabla}_{\boldsymbol{w}}\log p(\boldsymbol{w})] \tag{91}$$

$$\tilde{\boldsymbol{\nabla}}_{\boldsymbol{\Lambda}}L = -\mathbb{E}_q[\boldsymbol{\nabla}_{\boldsymbol{w}}^2 L(\boldsymbol{w}) - \rho\boldsymbol{\nabla}_{\boldsymbol{w}}^2 p(\boldsymbol{w})] + \rho\boldsymbol{\Lambda} \tag{92}$$

*Proof.* By Lemma 2, the gradients $\boldsymbol{\nabla}_{\phi'}$ of the objective $\mathcal{L}(\boldsymbol{\phi})$ w.r.t. $\phi' = \begin{bmatrix} \boldsymbol{\mu} \\ \mathrm{vec}(\boldsymbol{\Sigma}) \end{bmatrix}$ are given by:

$$\begin{aligned} \boldsymbol{\nabla}_{\boldsymbol{\mu}}L &= \mathbb{E}_q[\boldsymbol{\nabla}_{\boldsymbol{w}}L(\boldsymbol{w}) - \rho\boldsymbol{\nabla}_{\boldsymbol{w}}\log p(\boldsymbol{w})] \\ \boldsymbol{\nabla}_{\boldsymbol{\Sigma}}L &= \frac{1}{2}\mathbb{E}_q[\boldsymbol{\nabla}_{\boldsymbol{w}}^2 L(\boldsymbol{w}) - \rho\boldsymbol{\nabla}_w^2\log p(\boldsymbol{w})] - \frac{\rho}{2}\boldsymbol{\Sigma}^{-1} \end{aligned} \tag{93}$$

The Fisher Information Matrix is given by equation A.2:

$$F = \mathbb{E}_{q_{\phi}}\left[-\boldsymbol{\nabla}_{\phi}^2\log q_{\phi}\right] = \begin{bmatrix} \boldsymbol{\Sigma}^{-1} & 0 \\ 0 & \frac{1}{2}\boldsymbol{\Sigma}^{-1}\otimes\boldsymbol{\Sigma}^{-1} \end{bmatrix} \tag{94}$$

and therefore

$$F^{-1}\boldsymbol{\nabla}_{\boldsymbol{\phi}}L(\boldsymbol{\phi}) = \begin{bmatrix} \boldsymbol{\Sigma} & 0 \\ 0 & 2\boldsymbol{\Sigma}\otimes\boldsymbol{\Sigma} \end{bmatrix} \begin{bmatrix} \boldsymbol{\nabla}_{\boldsymbol{\mu}}L \\ \text{vec}(\boldsymbol{\nabla}_{\boldsymbol{\Lambda}}L) \end{bmatrix} = \begin{bmatrix} \boldsymbol{\Sigma}\boldsymbol{\nabla}_{\boldsymbol{\mu}}L \\ \text{vec}(2\boldsymbol{\Sigma}\boldsymbol{\nabla}_{\boldsymbol{\Lambda}}L\boldsymbol{\Sigma}) \end{bmatrix} \tag{95}$$

where we used the identities $(\boldsymbol{B}^{\boldsymbol{T}}\otimes\boldsymbol{A})\text{vec}(\boldsymbol{X}) = \text{vec}(\boldsymbol{A}\boldsymbol{X}\boldsymbol{B})$ and $(\boldsymbol{A}\otimes\boldsymbol{B})^{-1} = \boldsymbol{A}^{-1}\otimes\boldsymbol{B}^{-1}$. The gradient $\tilde{\boldsymbol{\nabla}}_{\boldsymbol{\mu}}L$ then follows immediately from the definition of the natural gradient operator. Using the chain rule for matrix derivatives we have that:

$$\boldsymbol{\nabla}_{\boldsymbol{\Lambda}}L = -\boldsymbol{\Lambda}\boldsymbol{\nabla}_{\boldsymbol{\Sigma}}\boldsymbol{\Lambda} \tag{96}$$

Since $\text{vec}(2\boldsymbol{\Sigma}\boldsymbol{\nabla}_{\boldsymbol{\Lambda}}L\boldsymbol{\Sigma}) = \text{vec}(-2\boldsymbol{\nabla}_{\boldsymbol{\Sigma}}L)$, we have the required updates. $\qquad\square$

## A.6 Convergence analysis

With $T(\boldsymbol{w}_t) := \left\langle (\boldsymbol{\nabla}_{\boldsymbol{w}}L(\boldsymbol{w}_t))^2, (\overline{\boldsymbol{f}}+\delta)_t^{-1} \right\rangle$, as $\rho \to 0$, the iterates $\boldsymbol{w}_{t+1} = \boldsymbol{w}_t - \alpha_t\boldsymbol{\nabla}_{\boldsymbol{w}}\left[L(\boldsymbol{w}_t)+\rho\boldsymbol{\nabla}_{\boldsymbol{w}}T(\boldsymbol{w}_t)\right]$ will converge to those of SGD. For $\rho > 0$, the algorithm is biased away from a pure descent direction, and convergence then depends on the magnitude of $\rho$. The key assumption in the following convergence proof is that $\|\rho\boldsymbol{\nabla}_{\boldsymbol{w}}T(\boldsymbol{w}_t)\|_2^2 \leq \kappa\|\boldsymbol{\nabla}_{\boldsymbol{w}}L(\boldsymbol{w}_t)\|_2^2 + \zeta$, which controls the bias. This follows from the standard assumption of twice-differentiability of $L(\boldsymbol{w})$ and the Lipschitz continuity of $\boldsymbol{\nabla}_{\boldsymbol{w}}L(\boldsymbol{w}_t)$, which imply that the Hessian has a bounded spectral norm:

$$\begin{aligned} \|\rho\boldsymbol{\nabla}_{\boldsymbol{w}}T(\boldsymbol{w}_t)\|_2^2 &\leq 4\rho^2\|\boldsymbol{\nabla}_{\boldsymbol{w}}^2 L(\boldsymbol{w}_t)\|_2^2\|(\overline{\boldsymbol{f}}+\delta)_t^{-1}\|_2^2 \\ &\leq 4\left(\frac{\rho}{\delta}\right)^2 C^2 p \end{aligned} \tag{97}$$

so that $\zeta$ depends on the Lipschitz constant $C$ and the ratio $\frac{\rho}{\delta}$.

**Theorem 3.** *Let* $T(\boldsymbol{w}_t) := \left\langle (\boldsymbol{\nabla}_{\boldsymbol{w}}L(\boldsymbol{w}_t))^2, \overline{\boldsymbol{f}}_t^{-1} \right\rangle$, *and assume the objective (loss)* $L : \mathbb{R}^p \to \mathbb{R}$ *is Lipschitz continuous, twice differentiable, and has Lipschitz-continuous gradient. Let us assume, following Bottou et al. (2016) and Ajalloeian & Stich (2021) that we have a stochastic direction* $g(\boldsymbol{w}_t, \boldsymbol{\xi}_t)$ *which has the following properties,* $\forall t$:

$$\mathbb{E}\left[g(\boldsymbol{w}_t, \boldsymbol{\xi}_t)\right] = \boldsymbol{\nabla}_{\boldsymbol{w}}L + \rho\boldsymbol{\nabla}_{\boldsymbol{w}}T(\boldsymbol{w}_t) \tag{98}$$

*and further assuming that there exist* $M, M_G$ *such that,* $\forall t$,

$$\mathbb{E}\left[\|g(\boldsymbol{w}_t, \boldsymbol{\xi}_t)\|^2\right] \leq M + M_G\|\boldsymbol{\nabla}_{\boldsymbol{w}}L + \rho\boldsymbol{\nabla}_{\boldsymbol{w}}T(\boldsymbol{w}_t)\|^2 \tag{99}$$

*and the following bound on the bias:*

$$\|\rho\boldsymbol{\nabla}_{\boldsymbol{w}}T(\boldsymbol{w}_t)\|^2 \leq \kappa\|\boldsymbol{\nabla}_{\boldsymbol{w}}L(\boldsymbol{w}_t)\|_2^2 + \zeta \tag{100}$$

*then the iteration:*

$$\begin{aligned} \boldsymbol{w}_{t+1} &= \boldsymbol{w}_t - \alpha_t\boldsymbol{\nabla}_{\boldsymbol{w}}\left[L(\boldsymbol{w}_t)+\rho\boldsymbol{\nabla}_{\boldsymbol{w}}T(\boldsymbol{w}_t)\right] \\ \overline{\boldsymbol{f}}_{t+1} &= (1-\beta)\overline{\boldsymbol{f}}_t + \beta\left(\boldsymbol{\nabla}_{\boldsymbol{w}}L(\boldsymbol{w}_t)\right)^2 \end{aligned} \tag{101}$$

*converges to a neighborhood of a stationary point with* $\|\boldsymbol{\nabla}L(\boldsymbol{w})\|_2^2 = \mathcal{O}(\zeta)$.

*Proof.* By the Lipschitz continuity of the objective function we have the quadratic bound:

$$L(\boldsymbol{y}) \leq L(\boldsymbol{x}) + \langle\boldsymbol{\nabla}_{\boldsymbol{w}}L(x), \boldsymbol{y}-\boldsymbol{x}\rangle + \frac{C}{2}\|\boldsymbol{y}-\boldsymbol{x}\|^2 \tag{102}$$

By the quadratic upper bound, the iterates generated by the algorithm satisfy:

$$L(\boldsymbol{w}_{t+1}) - L(\boldsymbol{w}_t) \leq -\alpha_t\langle\boldsymbol{\nabla}_{\boldsymbol{w}}L(\boldsymbol{w}_t), g(\boldsymbol{w}_k, \boldsymbol{\xi}_k)\rangle + \frac{1}{2}\alpha_t^2 C\|g(\boldsymbol{w}_k, \boldsymbol{\xi}_k)\|_2^2 \tag{103}$$

Taking expectations and applying the variance bound we have:

$$\mathbb{E}L(\boldsymbol{w}_{t+1}) - L(\boldsymbol{w}_t)$$

$$\leq -\alpha_t \|\boldsymbol{\nabla}_{\boldsymbol{w}} L(\boldsymbol{w}_t)\|^2 - \alpha_t \rho \boldsymbol{\nabla}_{\boldsymbol{w}} L(\boldsymbol{w}_t)^T \boldsymbol{\nabla}_{\boldsymbol{w}} T(\boldsymbol{w}_t) + \frac{1}{2}\alpha_t^2 C \mathbb{E}\left[\|g(\boldsymbol{w}_k, \boldsymbol{\xi}_k)\|_2^2\right]$$

$$= -\alpha_t \|\boldsymbol{\nabla}_{\boldsymbol{w}} L(\boldsymbol{w}_t)\|^2 - \alpha_t \rho \boldsymbol{\nabla}_{\boldsymbol{w}} L(\boldsymbol{w}_t)^T \boldsymbol{\nabla}_{\boldsymbol{w}} T(\boldsymbol{w}_t) + \frac{1}{2}\alpha_t^2 C \left[M + M_G \|\boldsymbol{\nabla}_{\boldsymbol{w}} L(x) + \rho \boldsymbol{\nabla}_{\boldsymbol{w}} T(\boldsymbol{w}_t)\|_2^2\right]$$

$$= -\alpha_t \|\boldsymbol{\nabla}_{\boldsymbol{w}} L(\boldsymbol{w}_t)\|^2 - \alpha_t (1 - \alpha C M_G) \rho \boldsymbol{\nabla}_{\boldsymbol{w}} L(\boldsymbol{w}_t)^T \boldsymbol{\nabla}_{\boldsymbol{w}} T(\boldsymbol{w}_t) + \frac{1}{2}\alpha_t^2 C M + \frac{1}{2}\alpha_t^2 C M_G \left(\|\boldsymbol{\nabla}_{\boldsymbol{w}} L(x)\|_2^2 + \rho\|\boldsymbol{\nabla}_{\boldsymbol{w}} T(\boldsymbol{w}_t)\|_2^2\right)$$

$$(104)$$

So that, choosing $\alpha_t < \frac{1}{CM_G}$ and applying the bound on $\|\boldsymbol{\nabla}_{\boldsymbol{w}} T(\boldsymbol{w}_t)\|$ we have:

$$\mathbb{E}L(\boldsymbol{w}_{t+1}) - L(\boldsymbol{w}_t) \leq -\frac{1}{2}\alpha_t \|\boldsymbol{\nabla}_{\boldsymbol{w}} L(\boldsymbol{w}_t)\|^2 + \frac{1}{2}\alpha_t^2 C M + \frac{1}{2}\alpha_t \|\rho \boldsymbol{\nabla}_{\boldsymbol{w}} T(\boldsymbol{w}_t)\|_2^2$$

$$\leq -\frac{1}{2}\alpha_t (1 - \kappa) \|\boldsymbol{\nabla}_{\boldsymbol{w}} L(\boldsymbol{w}_t)\|^2 + \frac{1}{2}\alpha_t^2 C M + \frac{\alpha_t}{2}\zeta$$

$$(105)$$

Taking the total expectation, for a fixed $\alpha$, we then have:

$$L_{inf} - L(\boldsymbol{w}_1) \leq \mathbb{E}\left[L(\boldsymbol{w}_{K+1})\right] - L(\boldsymbol{w}_1) \leq -\frac{1}{2}\alpha(1 - \kappa)\sum_{t=1}^{K} \|\boldsymbol{\nabla}_{\boldsymbol{w}} L(\boldsymbol{w}_t)\|^2 + \frac{1}{2}K\alpha^2 C M + \frac{K\alpha}{2}\zeta \qquad (106)$$

Finally, we have that:

$$\frac{1}{K}\sum_{t=1}^{K} \|\boldsymbol{\nabla}_{\boldsymbol{w}} L(\boldsymbol{w}_t)\|^2 = \frac{\alpha C M}{1 - \kappa} + 2\frac{F(\boldsymbol{w}_1) - F_{inf}}{K\alpha(1 - \kappa)} \xrightarrow{K \to \infty} \frac{\alpha C M}{1 - \kappa} + \frac{\zeta}{1 - \kappa} \qquad (107)$$

$$\square$$

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
