# OpenReview forum: "G-TRACER: Expected Sharpness Optimization"
_TMLR — Rejected by TMLR_

### Review · Reviewer_LcJi · 2024-05-31

**Summary Of Contributions:**

The authors propose a regularizer that promotes flatness and is derived from an ELBO-type objective. The proposed G-Tracer overperforms SAM in the challenging setting of large batch training. In an experiment, G-Tracer improves performance as the regularization increases whereas SAM crashes at zero accuracy for large $ \rho $.

**Audience:**

Yes

**Claims And Evidence:**

Yes

**Requested Changes:**

The exposition needs a major revision (see above).

**Strengths And Weaknesses:**

Looking for a theory-inspired regularizer is interesting broadly as it changes implicit bias and may promote desirable properties. However, unfortunately, the exposition of the paper makes it hard to understand the proposed method let alone connections with Fished information matrix (FIM), PAC-Bayes, and Variational Loss.

In the introduction, the interpolating solutions and the loss geometry of the connected global minima in this setting are emphasized but it is unclear how the proposed algorithm behaves near a connected global minima manifold and whether it picks the flatter minima therein (see Chaoyue Liu, Libin Zhu, and Mikhail Belkin).

In Algorithm 1, what does the square of the gradient mean? How is it divided by f which I think is another vector? Also, vectors are treated as scalars at other places, preventing me from being able to read the equations.

It is written several times that G can be interpreted as a Fisher matrix but it is unclear what G is until very late! Is G equivalent to Eq 2@ which is the inverse of the FIM?

The paper claims in the conclusion that this algo provably converges without any such proof. The only theorems presented in the paper are revisions from PAC-Bayes literature but it is unclear how these help to build intuition for G-Tracer.

For a paper that proposes no new theorem, the experimental evaluation is limited.

---

> ### Comment · Action_Editor_fX1c · 2024-06-06
> **Kindly Expand Your Review**
>
> Dear Reviewer,
>
> Thank you for submitting your review. We understand this is a busy season, and we greatly appreciate your time and effort.
>
> To help the authors best understand and apply your feedback, we kindly request that you expand your review slightly. A few additional sentences highlighting specific areas for improvement would be immensely helpful.
>
> Thank you again for your valuable contribution.
>
> Sincerely,
> AC

---

> > ### Comment · Reviewer_LcJi · 2024-06-13
> > **Review Expanded**
> >
> > Dear AC,
> >
> > Thank you for your kind note. I have expanded the review as prompted by your message. I hope this review will be valuable for evaluating this paper.
> >
> > Best regards,

---

> ### Author Response · Authors · 2024-07-03
> **Rebuttal**
>
> We thank the reviewer for their comments which we have fully taken on board.  In particular, we have added significantly to the "Problem Setting" section, as well as to the main derivations and proofs, to address the multiple confusions that are rightly referred to by the reviewer.  We believe that the paper has been greatly improved as a result.  Addressing the other points:
>
> 1. Regarding the "square of the gradient".
> There was an error in the initial version of the manuscript (where the dot product was omitted).  SGD-Tracer can be written as
> \begin{equation}
>     \rho\big(\mathbf{\nabla_w} L(\mathbf{w_t})\big)^2  \cdot 1/\mathbf{f_t} = \rho||\mathbf{\tilde{F}}^{-1}\mathbf{\nabla_w} L(\mathbf{w_t})||_2^2
> \end{equation}(where $\mathbf{\tilde{F}}$ is the diagonal matrix with diagonal given by $\sqrt{\mathbf{f_t}}$).
> We have clarified in the manuscript that operations are to be understood elementwise (other than the dot product).  The expression in the end is a convenient way to write the metric Hessian trace in the special case that both are diagonal.  In effect SGD-TRACER is a preconditioned gradient (squared-)norm penalty where the preconditioner is the same as used in eg Adam.
> 2. Regarding the use of $\mathbf{F}$  and $\mathbf{G}$ in the text.  This has been clarified and made fully consistent.  $\mathbf{G}$ is now explicitly introduced as the smoothed FIM, the inverse of which is the preconditioner.  $\mathbf{G}$ is used to remind/suggest the reader that we are dealing with a Riemannian metric.
> 3. The convergence proof is found in the Appendix and is referred to in the main body of the text.  We haven't explicitly labelled it as a Theorem, but we will consider doing this.
> 4. Regarding Theorems in general, the paper rigorously derives a novel, coordinate-free regularizer from general variational principles, shows rigorously how it is connected to SAM and other gradient penalties, rigorously proves convergence, and establishes a number of deep connections with geometrical operators and concepts.  Many of these results could be cast as Theorems, but we have preferred to keep the presentation and development of the theory as direct as possible.  Nonetheless, between the main text and Appendices, we do have 1 Theorem, 2 propositions and 2 lemmas.
> 5. For a theory-driven paper with rigorous results and derivations, we believe the experimental validations are in line with other leading papers in this area (see the papers cited in the Results section).  In particular, we have covered CV and NLP modalities, and both standard and challenging (eg noisy) problem variants.  We have also reproduced results from the benchmark papers in the area.
> 6. Finally, regarding the question of connected global minima.  This is an interesting question, and surely a worthwhile area for further study.  What we observe empirically is that the generalization performance of G-TRACER continues to improve even as the training error converges to zero, which is consistent with a penalty-boosting effect, n the neighborhood of a minimum, of the Adam-style preconditioner in the gradient norm.  Note that unlike SAM, this happens in anisotropic way which, given the large number of degenerate directions at a minimum, agrees with intuition.

---

### Review · Reviewer_Vcx7 · 2024-06-04

**Summary Of Contributions:**

In this work, the authors attempt to find a sharpness regularizer which is defined in a coordinate independent way. They introduce the G-tracer - a penalty based on the trace of the Hessian times the inverse of the fisher information matrix. The authors carry out a theoretical analysis to motivate a particular approximation of this quantity which ends up looking like the ratio of the individual per-coordinate gradient norms, divided by an exponential moving average of the same quantity over multiple batches. They then show that the g-tracer penalty shows improvements over SAM in certain noisy or hard vision tasks, and some NLP tasks as well.

**Audience:**

Yes

**Claims And Evidence:**

No

**Requested Changes:**

Overall I found the exposition in Section 2 to be very confusing - it was unclear what the settings were for computing the various forms of the Fisher information that show up in the motivation and derivations. This needs to be cleaned up and made much more explicit in order for me to recommend publication. On possible solution would be to invert the order of presentation, and start with SGD-tracer algorithm, and go through the steps/assumptions used to derive it.

More detailed comments:

For the line:

"Much of the literature on loss surface flatness and generalization has been concerned with non-geometric measures of sharpness and flatness"

I think non-geometric is the wrong way to describe this. Many geometric quantities can be written/defined in a particular coordinate system; it is then up to the geometer to use rules to transform these quantities to other coordinate systems. It seems that the authors real issue is with constructions that are not parameter-independent. I suggest that authors use this or some other modified language to make their (very good point) more precise.

In section 1.2.2, the authors make the connection to the Laplace-Beltrami operator. Would it be possible to write a short appendix which summarizes what utility Laplace-Beltrami has over the "standard" coordinatewise Laplacian? This request is optional, but since this work provides such a clear introduction of certain differential geometric quantities to the ML community, I think it would make the work even stronger as a reference.


Additional references:

In section 1.3.2 when discussing the related literature for SAM: an alternative analysis of SAM analyzes the update rule away from equilibrium, and provides one explanation for the batch-size dependence of the regularizer:

https://proceedings.mlr.press/v202/agarwala23a/agarwala23a.pdf

The final form of the gtracer is related to gradient penalty regularizers:

https://arxiv.org/abs/2009.11162
https://arxiv.org/abs/2202.03599

which can be considered an approximation to SAM:

https://arxiv.org/abs/2401.10809

**Strengths And Weaknesses:**

I thought the idea of trying to find parameterization-invariant sharpness measures/regularizers is a good one. I also think that the specific form found (SGD-tracer) is very simple and builds on previous sharpness works in an interesting way. The experimental results seem promising with some caveats (listed below). However, overall there are massive issues with the presentation which make it hard to judge the correctness and utility of the theoretical analysis.

In general, I found the presentation of the G-tracer, and particularly the various definitions of the fisher information, very confusing. I initially thought the Fisher information was coming from the fact that the model was inducing probabilities on its outputs via e.g. a softmax, for fixed parameters; however later sections seem to focus on Gaussian distributions of the parameters themselves, independent of any relationship to the model. In particular, I found section 2.1.2 to be very confusing, as the algorithm in (17) seems to make no reference to Gaussian distributions or $\mu$, while all the analysis in 2.1.2 does.

In the definition of the G-tracer, the Hessian is approximated via the Fisher information. Does this mean the approach only works for losses that can be written as log-likelihoods? For log-likelihood losses: is this construction the same as the Gauss-Newton part of the Hessian?

Another point that confuses me: if G is the Fisher Information, and H is also the Fisher Information, why does the G-tracer have non-trivial information? I understand that in the implemented algorithm, H is supposed to be the minibatch Fisher Information and G is an estimate of the full-batch Fisher - but in the limit of large batch sizes (or small learning rate), won't these coincide? This does not seem to solve the dependence on batch size noted as an issue with SAM.

In the definition of SGD tracer, sum of gradients squared is replaced with squared sum of gradients. It's unclear to me if this leads to changes in the magnitude/distribution of the tracer quantities, which lead it away from the original motivation.

Another overall point: the authors started off by criticizing other sharpness measures for not having sufficiently intrinsic definitions. However, the resulting algorithm also suffers from this issue; it's constructed out of the diagonals of two matrices, which induces a particular coordinate dependence. Also in practice the derivative is not taken through both parts of the ratio but only the numerator. These both greatly weaken the claim that the G-tracer method should be thought of as a coordinate-free, covariant metric.

The results in section 3.1.2 are interesting, but it's not immediately clear what to take away from them; it may indeed be the case that gtracer is less sensitive massive corruptions and lack of regularization compared to SAM, but it's unclear what the practical takeaways are. The ViT results are potentially more interesting, but those are hard to interpret as well. It would be stronger if there was evidence for gtracer improving over SAM in a setting where both are performing well overall on the task; to me the practical relevance of SAM is something that might take an already good model over the top, rather than taking a bad model and making it slightly less bad.

The results on NLP seem very promising, as those networks are trained in a more practical setting.

My main barrier to marking "Yes" on claims and evidence is the confusing state of the theory in Section 2; I look forward to reading other reviewers thoughts on this matter and a productive dialogue with the authors as well.

---

> ### Author Response · Authors · 2024-07-02
> **Rebuttal pt 1**
>
> We thank the reviewer for the thoughtful and constructive comments.  We are pleased that they find the idea to be interesting, that it builds on previous work in an interesting way, and that the experimental results are promising.
>
> Regarding the Requested Changes and major points:
>
> 1. We have fully taken on board the comments regarding the presentation and have inverted the order of presentation as the reviewer suggests.  We are very grateful for this suggestion and believe that the new manuscript is significantly clearer as a result.  We have also expanded the introductory "Problem setting" section to carefully and clearly explain the probabilistic (general variational inference) and point estimation setups, and corresponding FIMs and associated geometry.
> 2. Regarding the use of the term "Geometric" in connection with invariance, we agree and have changed the wording to consistently refer to constructions not depending on arbitrary parameterization.
> 3. We agree that a short primer on differential geometry covering the manifold Laplacian would be helpful.  We will include this in an Appendix and update the manuscript when available.
> 4. Additional references.  These have been extremely helpful, so much so that have added a section covering the connection with gradient penalty regularization and SAM.
>
> Regarding minor points:
> 1. Regarding The Fisher Approximation to the Hessian.  The GGN matrix and the Hessian do indeed coincide (see eg [1] and [2]) for the most practically relevant losses: cross-entropy (classification) and squared error (regression), corresponding to exponential family output distributions with natural parameters given by the output function $f(\mathbf{x},\mathbf{w})$.  In that case the log-likelihood FIM converges to the Hessian as the training error goes to zero, and is equivalent to the Generalized Gauss-Newton matrix.  In practice, the use of the FIM as a Hessian substitute for general losses is widespread and more loosely justfied by the fact that:
> \begin{equation}
> \begin{aligned}
>     \mathbf{F} & = \mathbb{E}[(\nabla \log p(\mathbf{y}| \mathbf{x}, \mathbf{w})) (\nabla \log p(\mathbf{y}| \mathbf{x}, \mathbf{w}))^T]
>     = \mathbb{E}\left[-\mathbf{\nabla^2}_{\mathbf{w}} \log p(\mathbf{y}| \mathbf{x}, \mathbf{w})\right]
> \end{aligned}
> \end{equation}
>
> 2. Regarding the question of how G-TRACER can have non-trivial information as a ratio of two related quantities: the matrix $\mathbf{G}$ plays the role of a metric in the metric trace expression in the update equation for the parameters, and is a constant for that update.  To take a concrete example, in the  in the limiting case of no smoothing in the update equation for $G$, as shown in the new section on gradient penalties, the effect of $G$ is to adjust the hessian trace to account for the local geometry in a batch just as it does in the full-batch case.  Concretely, SGD-TRACER can be written as
> \begin{equation}
>     \rho\big(\mathbf{\nabla_w} L(\mathbf{w_t})\big)^2  \cdot 1/\mathbf{f_t} = \rho||\mathbf{\tilde{F}}^{-1}\mathbf{\nabla_w} L(\mathbf{w_t})||_2^2
> \end{equation}(where $\mathbf{\tilde{F}}$ is the (constant) diagonal matrix with diagonal given by $\sqrt{\mathbf{f_t}}$) so that the gradient norm penalty is scaled in the same way as the gradients in adaptive optimizers (Adam, AdaGrad, RMSProp).  Differentiating through this penalty gives you a HVP as outlined in our rebuttal to K7N4's review.
>
> 3. Regarding the use of the gradient magnitude approximation.   The theoretical justification for this is given in Theorem 1 of [3] who show that the sum of squared gradients is close to the squared sum of gradients if the batch and population estimates are sufficiently close.  While this is standard and extremely widely used (for example in Adam, AdaGrad and RMSProp), it is of course biased, and there is scope for improvement here.  Computation of per-example gradients is in principle no more expensive, however, current autodiff libraries do not support it out of the box, so we opted for a simple, practical solution here.
>
> 4. Regarding the coordinate dependence of the mean-field approximations used in SGD-TRACER.  While SGD-TRACER is invariant to coordinate rescaling, it is absolutely the case that zeroing off-diagonal elements is a coordinate-dependent move.  As with equivalent mean-field constructions (eg Adam) the goal is to balance ease of implementation and tractability.  We plan to investigate non-diagonal alternatives such as KFAC  in future work.

---

> > ### Author Response · Authors · 2024-07-02
> > **Rebuttal pt 2**
> >
> > 5. Regarding results.  We are glad that the reviewer found the NLP results promising.  Regarding the CV results, we show, for example, on CIFAR-100/ResNet-20 where SAM shows an improvement in line with the results in the original paper (as well as follow-up papers, referenced in the text), that our method is competitive and consistent with other results in the SAM literature on SAM and its variants.  The most striking results in the SAM paper are the performance on challenging/noisy problems where performance improves not by fractions of a percent, but by an order of magnitude more than that.  For us the key practical relevance of these methods is not to make marginal improvements that are small compared to the gains made by, eg data augmentation or pre-training, but rather to develop methodology that can succeed on challenging problems that practitioners encounter in the real world.  Consider, for example, financial time series prediction, where there is a lack of symmetries/invariants so that augmentation is hard/impossible, effective sample sizes are small, and the signal-to-noise ratio is extremely low.  In this and many other real-world settings, it is often not that case that any method (eg SGD or SAM) is bad per se, rather that the problems are inherently challenging.  Having noted that the really striking results in the original SAM paper are the ones relating to challenging problem variants (eg noisy CIFAR) we therefore included such problems in our benchmarking.  We plan to benchmark our method on challenging, nonstationary, noisy time series data in future work.
> >
> > [1] Limitations of the Empirical Fisher Approximation, Kunstner et al., NeurIPS, 2019
> >
> > [2] Martens, New Insights and Perspectives on the Natural Gradient Method, JMLR, 2020
> >
> > [3] Khan et al., Fast and Scalable Bayesian Deep Learning by Weight-Perturbation in Adam, ICML, 2018

---

> > > ### Comment · Reviewer_Vcx7 · 2024-07-09
> > > **Response to changes**
> > >
> > > I first thank the authors for their re-working of the text; I find it overall much easier to read. In addition, the clarification that the new quantity is invariant to affine transformations in particular makes the picture much more precise and much clearer.
> > >
> > > "GGN matrix and the Hessian do indeed coincide" I do want to point out that this is only true at interpolating minima; much of training happens outside of this regime!
> > >
> > > It is also mentioned that the g-tracer form is invariant near critical points of the loss. Is this because the input to the trace is the identity, since Hessian becomes the GN? If so, this is 1. true only at an interpolating minimum, where the gradient is 0 for all datapoints and 2. does not seem like a good motivation for the method. I would appreciate some additional clarity on this point.
> > >
> > > Overall, I still find confusing how the theory ends up with minibatch GN times the inverse of (an estimate of) the full batch Gauss Newton, without implying that this construction will approach the identity matrix in many settings (including those where the batch size is large, a setting where SAM is said to be weak). I appreciate the clarification that only the "numerator" is differentiated through, which gives a non-trivial change to the update rule; however the overall motivation still feels weak.
> > >
> > > Regarding point 5 in the rebuttal: I think the paper would be much improved with some of these real challenging examples in hand; I appreciate the inspiration from the original SAM paper, but to my mind the positive reception of SAM was most bouyed by its improvement on ImageNet.
> > >
> > > Small additional notes:
> > >
> > > The reference
> > >
> > > https://arxiv.org/abs/2401.10809
> > >
> > > also studies Gauss-Newton trace penalties explicitly, and finds them to be comparable to SAM on some tasks - which I believe supports this paper's comparison of G-tracer as a "geometry aware SAM".
> > >
> > > For the definition of g-tracer, should the inverse really be a pseudo-inverse since G can have 0 eigenvalues? I know in practice you use a damping factor, but at least in the theoretical analysis maybe psuedo-inverse better captures the behavior?

---

### Review · Reviewer_K7N4 · 2024-06-16

**Summary Of Contributions:**

This paper proposes a new regularization term for optimization. When applied to SGD, the proposed update rule is:

$$
w_{t+1}=w_t-\alpha_t\nabla \Big(L(w_t)+\rho\big(\nabla L(w_t)\big)^2/f_t\Big),
$$

where $w_t$ is the model weight, $L(w_t)$ is the loss function (evaluated on a single batch), and $f_t$ is a running-average of $\big(\nabla L(w_t)\big)^2$ (i.e. gradient squares). So the proposed regularization term above is:

$$
\rho\nabla\big(\nabla L(w_t)\big)^2/f_t.
$$

The authors theoretically link this term to the sharpness of loss landscape, emphasizing its geometric meaning and reparameterization invariance. The reparameterization invariance is quite obvious here, because $f_t$ is a running-average of $\big(\nabla L(w_t)\big)^2$, so the expectation of $\big(\nabla L(w_t)\big)^2/f_t$ is 1 (which does not depend on reparameterization of $w_t$).

Then, empirically the authors evaluated the proposed optimizer, compared with SAM, on CIFAR-100 and its noisy variants, and BERT fine-tuning on 3 tasks in SuperGLUE.

**Audience:**

Yes

**Broader Impact Concerns:**

None.

**Claims And Evidence:**

No

**Requested Changes:**

Please address my concerns listed above. Especially, if the authors still believe that the proposed method is indeed sharpness-aware optimization, please directly rebut me.

I'm not saying that the authors' derivation is unacceptable; my point is: (1) there is an alternative, simpler explanation; and (2) when you have some leaps in the theory (like the one using stochastic single-batch evaluation instead of expectation on whole data distribution, as I described above), we will need some experiments to verify that the proposed method is actually doing what we expect it to do, namely regularizing sharpness in this case. Such verification experiments should be conducted very carefully: because stochastic gradient noise, momentum, and L2 regularization etc. could all be strong forces that affect the optimization behavior around a critical point. One has to eliminate all these factors and prove that the proposed term actually regularizes sharpness.

**Strengths And Weaknesses:**

Strengths:

* The authors are certainly knowledgeable regarding the topic of sharpness-aware optimization. I learned a lot reading through the extensive literature review and the cited papers.

* Intuitively, sharpness regularization is closely related to the intrinsic geometry of loss landscape, and to bring the Fisher Information Matrix into the equation is an intriguing idea. The research direction is worth exploring.

Weakness:

However, specific to this work: currently I don't buy the story. Because if we look more closely at the proposed update rule; since

$$
\nabla\big(\nabla L(w_t)\big)^2=2\nabla L(w_t)\cdot\nabla\nabla L(w_t)
$$

has a gradient factor $\nabla L(w_t)$, we can just take it out and rewrite the update rule as:

$$
w_{t+1}=w_t-\alpha_t\nabla \Big(L(w_t)+\rho\big(\nabla L(w_t)\big)^2/f_t\Big)=w_t-\Big(\alpha_t+2\rho\nabla\nabla L(w_t)/f_t\Big)\nabla L(w_t).
$$

So a simpler explanation of the proposed SGD-TRACER algorithm, is that this is just SGD with some kind of adaptive learning-rate.

* Especially, when the optimization reaches a critical point which means $\nabla L(w_t)=0$, the update rule does nothing. No matter how you relate the formula to the geometry of loss landscape, this behavior just doesn't look like sharpness-aware optimization.

* Specific to the derivation in this paper: the one step I found most questionable is to use Fisher Information Matrix to replace Hessian. Indeed, we have the equality

$$
\mathbb{E}\Big[\big(\frac{\partial}{\partial w}L(w)\big)^2\Big]=-\mathbb{E}\Big[\frac{\partial^2}{\partial w^2}L(w)\Big],
$$

but this only holds when you take the expectation over the whole data distribution. Things become largely different when you evaluate these quantities on single batches in a stochastic optimization manner. I think Fisher Information Matrix is a good idea because intuitively it should be related to the sharpness of loss landscape; but I feel unsatisfied in the way it is used in this paper.

* More fundamentally, I feel that once you use second order approximation (Eq. 21) to analyze the behavior in a small neighborhood, the derivation will eventually end up at something similar to 2nd-order optimization rather than sharpness-aware optimization. SAM is interesting because it actually calculates the gradient on a small perturbation of the model parameter.

* Regarding the empirical evaluations: I'm not extremely familiar with the noisy settings of CIFAR-100; but I feel the performance gap reported in Table 5 is well within the random variance one commonly observes on fine-tuning BERT on SuperGLUE with Adam.

Due to these concerns listed above, I feel I should say "No" to the question "Are the claims made in the submission supported by accurate, convincing and clear evidence?".

---

> ### Author Response · Authors · 2024-07-02
> **Rebuttal pt 1**
>
> We thank the reviewer for the extremely helpful and insightful comments and are pleased that they found our paper both instructive and intriguing.  We have updated and improved the manuscript significantly as a result.
>
> Directly addressing the concerns in "Requested Changes":
>
> Regarding the reviewer's rewrite of the update rule and the connection with SAM: some confusion might have been caused here by a notational error in the initial version of the manuscript (where the dot product was omitted).  SGD-Tracer can be written as
> \begin{equation}
>     \rho\big(\mathbf{\nabla_w} L(\mathbf{w_t})\big)^2  \cdot 1/\mathbf{f_t} = \rho||\mathbf{\tilde{F}}^{-1}\mathbf{\nabla_w} L(\mathbf{w_t})||_2^2
> \end{equation}(where $\mathbf{\tilde{F}}$ is the diagonal matrix with diagonal given by $\sqrt{\mathbf{f_t}}$).
>
> The connection with sharpness aware optimization in the sense of SAM and its variants is readily shown, and we thank the reviewer for helping us to bring this to the fore, and thus to highlight the connections to literature on gradient penalties (also pointed out by reviewer Vcx7).  We have updated the manuscript to highlight these connections and believe that the paper is much improved as a result.  In summary, starting with the SGD-TRACER penalty and taking the gradient of the resulting augmented loss function we obtain:
> \begin{equation}
> L^{\mathcal{G}}(\mathbf{w_t}) = L(\mathbf{w_t}) + \rho||\mathbf{\tilde{F}}^{-1}\mathbf{\nabla_w} L(\mathbf{w_t})||_2^2
> \end{equation}
> and choosing $\beta = 1$ in the update equation for $\mathbf{f_t}$ (corresponding to no smoothing) we have:
> \begin{equation}
>    \mathbf{\nabla_w} L^{\mathcal{G}}(\mathbf{w_t}) = \mathbf{\nabla_w} L(\mathbf{w_t}) +  2 \rho \mathbf{F}^{-1} \mathbf{H} \mathbf{\nabla_w} L(\mathbf{w_t})
> \end{equation}
>
> Writing the Hessian-vector product (to second order) as
> \begin{equation}
> \mathbf{H} \mathbf{\nabla_w} L(\mathbf{w_t}) \approx \frac{\mathbf{\nabla_w} L(\mathbf{w_t} + \delta \mathbf{\nabla_w} L(\mathbf{w_t})) - \mathbf{\nabla_w} L(\mathbf{w_t})}{\delta}
> \end{equation} we get the following expression for the gradient of the penalized loss:
> \begin{equation}
> \begin{split}
>      \mathbf{\nabla_w} L^{\mathcal{G}}(\mathbf{w_t}) = \mathbf{\nabla_w} L(\mathbf{w_t}) +  2 \rho \mathbf{F}^{-1} \frac{\mathbf{\nabla_w} L(\mathbf{w_t} + \delta  \mathbf{\nabla_w} L(\mathbf{w_t})) - \mathbf{\nabla_w} L(\mathbf{w_t})}{\delta} \\
>      = (\mathbf{I}-\mathbf{\Gamma}) \mathbf{\nabla_w} L(\mathbf{w_t}) + \mathbf{\Gamma} \mathbf{\nabla_w} L(\mathbf{w_t} + \delta  \mathbf{\nabla_w} L(\mathbf{w_t}))
> \end{split}
> \end{equation}
> where $\Gamma :=2 \frac{\rho}{\delta} \mathbf{F}^{-1}$.
>
> Following the derivation of SAM, by dropping higher-order terms, we approximate:
>
> $$\mathbf{\nabla_w} L(\mathbf{w_t} + \delta \mathbf{\nabla_w} L(\mathbf{w_t})) \approx \mathbf{\nabla_w} L(\mathbf{w}) |_{\mathbf{w} = \mathbf{w_t} +  \delta  \mathbf{\nabla_w} L(\mathbf{w_t})}$$
>
> obtaining:
>
> \begin{equation}
> \begin{aligned}
>      \mathbf{\nabla_w} L^{\mathcal{G}}(\mathbf{w_t}) =(\mathbf{I}-\mathbf{\Gamma}) \mathbf{\nabla_w} L(\mathbf{w_t}) + \mathbf{\Gamma} \mathbf{\nabla_w} L(\mathbf{w}) |_{\mathbf{w} = \mathbf{w_t} +  \delta  \mathbf{\nabla\_{\mathbf{w}}} L(\mathbf{w_t})}
> \end{aligned}
> \end{equation}
>
> The second term is proportional to $\mathbf{F^{-1}} \mathbf{\nabla_{\mathbf{w}}} L(\mathbf{w}) |_{\mathbf{w} = \mathbf{w_t} + \delta  \mathbf{\nabla\_{\mathbf{w}}} L(\mathbf{w_t})}$ which is almost identical to the SAM gradient update (in effect, a natural SAM gradient).  In the special case that we choose $\mathbf{F} = \mathbf{I}$ and $\mathbf{\Gamma} = \mathbf{I}$ we recover effectively the same update as SAM:
>
> \begin{equation}
> \begin{aligned}
>      \mathbf{\nabla_w} L^{\mathcal{G}}(\mathbf{w_t}) = \mathbf{\nabla_w} L(\mathbf{w}) |_{\mathbf{w} = \mathbf{w_t} +  \delta  \mathbf{\nabla\_{\mathbf{w}}} L(\mathbf{w_t})}
> \end{aligned}
> \end{equation}
>
> which is exactly the form used in many theoretical works and by practitioners (see eg [4], the canonical reference on the properties of SAM and the first proof of convergence, who show that the normalized perturbation is not important for generalization, and [5] as suggested by reviewer Vcx7).  Note that, to recover the the scaled unit-norm perturbation $ \mathbf{\nabla_w} L(\mathbf{w}) |_{\mathbf{w} = \mathbf{w_t} +  \delta  \frac{\mathbf{\nabla\_{\mathbf{w}}} L(\mathbf{w_t})}{||\mathbf{\nabla\_{\mathbf{w}}} L(\mathbf{w_t})||_2}}$ from the original SAM paper would require a 2-norm gradient penalty: $\rho||\mathbf{\nabla_w} L(\mathbf{w_t})||_2$.  The corresponding augmented loss has the same solution set as the squared 2-norm penalty, though the optimization dynamics are, of course, different.

---

> ### Author Response · Authors · 2024-07-02
> **Rebuttal pt 2**
>
> Thus, SAM can be seen as a special case of our more general scheme (we draw your attention to equation (18) in the modified manuscript for most general parameter-space - as opposed to variational parameter space - expression of our scheme), corresponding to $\beta=1$ (no smoothing), approximating an HVP, approximating the gradient of the resulting perturbed loss, and choosing as a preconditioner the identity matrix, so that the perturbation is not aligned with the natural geometry of the parameter space.  Notably, at a critical point, the normalized SAM update isn't well defined since the perturbation is undefined.  Whereas the normalized SAM perturbation radius is fixed, as our scheme approaches a critical point, the inverse Fisher has the effect of increasing the effective penalty in the directions where the gradient magnitudes are small.
>
> Our scheme shows how gradient-norm penalties (see [1]) can be derived from probabilistic principles.  In particular, natural-gradient norm penalization can thus be seen as a way to perform approximate variational inference.  G-TRACER in its simplest form, SGD-TRACER is thus not to be understood as an adaptive learning rate but rather as gradient norm penalization where the gradient is scaled by the inverse square root of the diagonal empirical Fisher.  This is exactly the gradient scaling used by adaptive optimziers such as Ada and RMSProp.  We also note that SGD-Tracer is merely a very special case of the general regularization scheme
> \begin{equation}
> \begin{aligned}
> \mathbf{w} \xleftarrow[]{} \mathbf{w} - \alpha \mathbf{\nabla}_{\mathbf{w}} [L(\mathbf{w})  +  \rho \text{Tr}(\mathbf{G^{-1}} \mathbf{H(w)})]\\\\
> \mathbf{G} \xleftarrow[]{} (1-\beta)\mathbf{G} + \beta \mathbf{F}
> \end{aligned}
> \end{equation}
> that we introduce, which is capable of being realized in many different ways.
>
> Another complementary perspective on the relationship between a penalty of this form is that the supremum of the gradient norm of a real-valued locally Lipschitz-continuous function is the Lipschitz constant which controls the regularity of the function, and in particular bounds the change (in norm) of the output for a given change in the input (see, eg [1]).
>
> We therefore have a multitude of related and complementary views on flatness arising from this formulation, including:
>
> 1) Metric trace, or Laplace-Beltrami, which gives the difference between the mean value of a function and the value of a function at a local minimum
> 2) Lipschitz constant control and thus regularity control
> 3) Gradient-norm penalty, adjusting for the natural geometry of the parameter space
> 4) Smoothing the loss by convolving with a Gaussian kernel, estimated from the data
>
> Regarding the use of the FIM in place of the Hessian: while the equality of FIM and the expected log-likelihood Hessian establish a connection between the FIM and the Hessian, we are not using large-sample asymptotics to justify the substitution.  The use of the FIM in place of the Hessian (in mini-batch settings) is standard and well-founded in approximation theory (see, eg [2] and [3] for an authoritative review) and the justification is that the FIM is equivalent to the Generalized Gauss-Newton matrix for exponential families (as is the case here in the log-likelihood FIM setting) and we have the following theorem [2]:
> \begin{equation}
>     ||\mathbf{\nabla^2} L(\mathbf{w}) - \textbf{G(w)} ||_2^2 < r(\mathbf{w})\beta
> \end{equation}
> where $r(\mathbf{w})$ are residuals that go to zero as the data are perfectly fit, $\mathbf{G}$ is the Generalized Gauss-Newton matrix, and $\beta$ is related to the smoothness of the coordinate functions of the neural network.  The limit here is not statistical, but rather approximation theoretic, and holds independently of batch size.  In connection with this, in response to the comments from all reviewers, we have reworked the derivations in the manuscript to highlight that the choices of Fisher and/or diagonal Fisher and empirical Fisher aren't specific to the overall method, as given by the above general update equations.
>
> Finally, on the minor points.  Regarding fine-tuning BERT: our experience is that results in the literature are indeed very hard to reproduce - for this reason, all results are reported with standard errors and represent averages of multiple separate runs.
>
> [1] Penalizing Gradient Norm for Efficiently Improving Generalization in Deep Learning, Yang et al.,  Proceedings of Machine Learning Research, 2022
>
> [2] Limitations of the Empirical Fisher Approximation, Kunstner et al., NeurIPS, 2019
>
> [3] Martens, New Insights and Perspectives on the Natural Gradient Method, JMLR, 2020
>
> [4] Andriushchenko, et al., Towards Understanding Sharpness-Aware Minimization ICML, 2022
>
> [5] SAM operates far from home: eigenvalue regularization as a dynamical phenomenon, Agarwala et al., 2023

---

> ### Comment · Reviewer_K7N4 · 2024-07-03
> **Thanks for the rebuttal**
>
> > Regarding the reviewer's rewrite of the update rule and the connection with SAM: some confusion might have been caused here by a notational error in the initial version of the manuscript (where the dot product was omitted).
>
> Yes I thought there might be a dot product; this does not change the situation. Let's write the SGD-Tracer regularization term in its entries:
> \begin{equation}
>     \rho\big(\mathbf{\nabla_w} L(\mathbf{w_t})\big)^2  \cdot 1/\mathbf{f_t} = \sum_i\frac{\rho}{f_i}g_i^2
> \end{equation}
> where we put $(f_i)=\mathbf{f_t}$ and $(g_i)=\mathbf{\nabla_w} L(\mathbf{w_t})$. When taking the gradient of this regularization term, $(f_i)$ is treated as constant, so
> $$
> \partial_j\big( \sum_i\frac{\rho}{f_i}g_i^2 \big) = \sum_i \frac{2\rho}{f_i}(\partial_j g_i)g_i,
> $$
> there is still a $g_i$ factor, although $i$ is summed across all suffices -- so this is a pre-condition matrix applied to the gradient, rather than element-wise learning-rate. I say this doesn't change the situation, because when $\mathbf{\nabla_w} L(\mathbf{w_t})=\mathbf{0}$, the SGD-Tracer regularization term still does nothing.
>
> It is true that SAM is not well-defined at a critical point; this is because SAM uses first-order approximation to calculate the perturbation $\mathbf{\epsilon}$, and that doesn't work when gradient is exactly 0 -- depending on the actual implementation, SAM might as well do nothing at a critical point. However, as SAM approaches (but not exactly at) a critical point, the perturbation $\mathbf{\epsilon}$ is always of fixed length $\rho$; in contrast, the perturbation in this work approaches 0 when it approaches a critical point. This is why I say the behavior does not look like sharpness-aware optimization.
>
> The authors might believe that the perturbation approaching 0 when approaching a critical point is a more reasonable behavior; I don't have a strong opinion either way. But as I described previously, this is the normal behavior of SGD with a pre-condition matrix; it is stretchy to frame it as a sharpness-aware optimization method.
>
> More specific to the derivation posted to the rebuttal:
> So you take the approximation at this step
> \begin{equation}
> \mathbf{H} \mathbf{\nabla_w} L(\mathbf{w_t}) \approx \frac{\mathbf{\nabla_w} L(\mathbf{w_t} + \delta \mathbf{\nabla_w} L(\mathbf{w_t})) - \mathbf{\nabla_w} L(\mathbf{w_t})}{\delta}
> \end{equation}
> which suggests $\delta$ should be sufficiently small; but then you absorb a $\frac{1}{\delta}$ factor into $\Gamma :=2 \frac{\rho}{\delta} \mathbf{F}^{-1}$ and "choose" $\mathbf{F} = \mathbf{I}$ and $\mathbf{\Gamma} = \mathbf{I}$, which means $\delta$ should be exactly $2\rho$, a constant.
>
> The behavior of SAM is like keeping $\rho$ to a constant; and the behavior of this work is like making $\delta$ approach 0. You cannot make $\delta$ approach 0 and equal $2\rho$ at the same time, and say "Thus, SAM can be seen as a special case of our more general scheme". It is just stretchy.

---

> > ### Author Response · Authors · 2024-07-03
> >
> > Thanks very much for the further comments.  We appreciate the time and thought and are grateful to have the tires kicked.
> >
> > So: it seems we agree that our method (in its simplest form, SGD-TRACER) is a preconditioned gradient penalty and not an adaptive learning rate.  We also agree that the behavior of SAM at a minimum is undefined.
> >
> > There are then two key things we still need to clarify and correct here:
> >
> > **SAM is equivalent to gradient-norm penalty**:  this is well known (see [1], section 3.3, also referenced by reviewer Vxc7) and the proof is exactly analogous to the derivation in our rebuttal.  In their derivation, as in ours, the choice $\delta \sim \rho$ (both constants) results in SAM.  It is not the case that $\delta \rightarrow 0$, rather simply that the approximation in the derivation is valid for small $\delta$.  So the point is that, so long as the perturbation $\rho$ is small (which it always is in practice), then SAM is equivalent to a gradient-norm penalty, since the choice (in their case) $\delta = \rho$ gives a valid approximation.
> >
> > Your claim that **SAM's generalization ability is related to the normalization and the unnormalized SAM is no different in effect to (preconditioned) SGD**
> >
> > First: we seem to be at cross-purposes here.  Much of the literature on SAM concerns exactly this unnormalized formulation.  The key reference is [2] who show experimentally that unnormalized vs normalized makes no difference to generalization.  We have also verfied this empirically on numerous occasions.  We have found unnormalized SAM to be harder to tune, though OOS gains are essentially identical to SAM (on average) with proper tuning.
> >
> > To put this matter to rest, reviewer Vxc7 helpfully refers to [4] by one of the lead/co-authors of the SAM paper from which we quote:
> >
> > - "We define a gradient penalty as an additive regularizer of the form: $L_{\mathrm{pen},p} = \rho ||\nabla L||^p$. Gradient penalties have recently gained popularity as regularizers (Barrett \& Dherin, 2021; Smith et al., 2021; Du et al., 2022; Zhao et al., 2022; Reizinger \& Huszar, 2023); this is in part due to their ability to reduce sharpness. In fact $L_{\mathrm{pen},p}$, is closely related to Sharpness Aware Minimization (SAM) (Foret et al., 2020). p = 1 corresponds to the original normalized formulation, **while p = 2 corresponds to the unnormalized formulation which is equally effective and easier to analyze** (Andriushchenko \& Flammarion, 2022; Agarwala \& Dauphin, 2023)."
> >
> > The choice p=2 is $exactly$ the special case of our algorithm we derived in our original rebuttal.  In general our normed gradient is of course preconditioned by the inverse FIM, which of course **does** have the effect of increasing penalization as a minimum is approached, and crucially, unlike SAM, in an anisotropic way.
> >
> > Another important recent reference is [4] who show in Theorem 3.2 that the **dynamics of unnormalized SAM correspond to SGD with an additional term that depends on the curvature of the loss**.  This directly contradicts the claim that the normalization is somehow responsible for the non-trivial dynamics (vs preconditioned SGD).
> >
> > Finally, while we can recover SAM by choosing an isotropic covariance $\sigma \mathbf{I}$ which exactly yields  $L_{\mathrm{pen},2} = \rho ||\nabla L||^2$, it is not a goal of this paper to simply perform "Sharpness Aware" optimization.  Our goal, clearly stated in the paper, is to replace the crude upper bound on $\mathbf{E_q} [L(w)]$ with an approximation to that integral over a region determined from the data, in a way that aligns with the geometry of the parameter space, and captures average flatness, and not "worst-case" sharpness.   The fact that both methods derive from the same variational/Bayes objective is a strong hint that they are related.  SAM bounds the expectation using a tail bound for $\chi$-squared RVs and the result is a Euclidian norm-ball perturbation.  Our method estimates the integral, capturing mean curvature, and because we are careful to optimize using natural gradients, we end up with a coordinate free method which adapts to the changing geometry of the parameter space.  In its most basic version, SGD-TRACER, our method is a preconditioned gradient norm (preconditioned in the same way that Adam is - by the sqrt of the EF).
> >
> > In order to propose a constructive way forward, we are very happy to qualify the claim that our method generalizes SAM with a discussion covering the above in more detail.  Please let us know if that is acceptable to you.  Thanks again for your thoughtful comments.
> >
> > [1] Penalizing Gradient Norm for Efficiently Improving Generalization in Deep Learning, Yang et al.,  Proceedings of Machine Learning Research, 2022
> >
> > [2] Andriushchenko, et al., Towards Understanding Sharpness-Aware Minimization ICML, 2022
> >
> > [3] Kim, et al., Stability analysis of sharpness-aware minimization, 2023
> >
> > [4] Compagnoni, et al., An SDE for Modeling SAM: Theory and Insights, 2023

---

> > > ### Comment · Reviewer_K7N4 · 2024-07-03
> > > **Further discussion**
> > >
> > > > SAM is equivalent to gradient-norm penalty: this is well known (see [Yang et al., 2022]) and the proof is exactly analogous to the derivation in our rebuttal.
> > >
> > > Yang et al. (2022) is indeed a generalization of SAM **because it actually implements the approximation step** -- the motivation of Yang et al. (2022) originates from gradient-norm penalty, and in order to avoid the calculation of Hessian, Yang et al. propose to approximate the Hessian by this
> > > \begin{equation}
> > > \mathbf{H} \mathbf{\nabla_w} L(\mathbf{w_t}) \approx \frac{\mathbf{\nabla_w} L(\mathbf{w_t} + \delta \mathbf{\nabla_w} L(\mathbf{w_t})) - \mathbf{\nabla_w} L(\mathbf{w_t})}{\delta}
> > > \end{equation}
> > > which then they show leads to a generalization of SAM. But this chain of reasoning becomes stretchy when applied to the story of this paper, **because G-Tracer is implementing the gradient-norm penalty without the approximation step**. Given the presentation of Algorithm 1 in this paper, I assume it just directly calculates the gradient of the gradient-norm penalty term -- which leads to an implicit calculation of the Hessian (it is a bit suspicious here why this is possible and the authors do not seem to face scalability issues, maybe there are extra tricks; but this is what it presents). So, G-Tracer becomes a pre-conditioned SGD, and the pre-condition matrix depends on the Hessian -- which makes G-Tracer closer to a 2nd-order optimizer rather than SAM. You could say that the way G-Tracer uses Hessian originates from the sharpness penalty and gradient-norm penalty, and an **approximation** of G-Tracer actually leads to a generalization of SAM; but you should **position G-Tracer as a 2nd-order pre-conditioned optimizer and also empirically compare to a bunch of 2nd-order optimizers**.
> > >
> > > As I said in the review, I'm not saying the derivation is unacceptable; but you are using an approximation in the derivation that is not implemented in the proposed algorithm, and currently that derivation is the only justification of the algorithm -- this is what I call stretchy.
> > >
> > > > Your claim that SAM's generalization ability is related to the normalization and the unnormalized SAM is no different in effect to (preconditioned) SGD
> > >
> > > I don't claim that; I don't have a strong opinion toward either normalized or unnormalized SAM. I think G-Tracer could work; it might work quite well. I just don't buy the story.
> > >
> > > Anyway, thanks to the authors for the discussion; I have learned a lot.
> > >
> > > I think preconditioned SGD, Hessian and 2nd-order optimization, sharpness and gradient-norm penalty, are certainly closely related, intriguing topics; but I think the story of this work should be thoroughly reconstructed, empirical comparisons (especially with 2nd-order optimization) should be added, in order for the paper to go through another round of review.

---

> ### Author Response · Authors · 2024-07-08
>
> Thanks for the engaging and very useful discussion.  To summarize the final position we have arrived at: you are saying that what is "stretchy" is, in the final SGD-TRACER step (i.e. in the special case that we assume a diagonal empirical Fisher approximation to the Hessian), not using the finite-difference approximation of the Hessian-vector product (HVP) $\mathbf{H} \mathbf{\nabla} L$.
>
> To this we make the following points.
>
> 1. Following this line of argument, we therefore agree that, if we use a finite difference HVP approximation, then our method does indeed generalize SAM, since it then generalizes Yang [1], and we agree Yang generalizes SAM.  In fact our implementation, for efficiency, uses exactly this finite difference approximation to the HVP, using the default guided by the recommendation of Andrei [4] to avoid numerical precision issues: $\delta = 2\sqrt{\epsilon} \frac{1 + ||\mathbf{w}||}{||\mathbf{\nabla}L||}$, where $\epsilon$ is machine precision.  For this choice, we have checked that the results are no different to double backpropagation (taking gradients through the gradient norm).
>
>    None of these approximations are derived from theory; rather, they arise from simplifications made for computational efficiency and while, for larger $\delta$ there are of course differences between the exact HVP and the finite-difference approximation (for smaller $\delta$ they are of course essentially identical, see below), the theory and empirical evidence point to this not being relevant to the fundamental connections between flatness and generalization.  Our own experiments confirm this, but see also eg, Table 1, from https://openreview.net/pdf?id=Gl4AsqInti [5] by one of the lead/co-authors of the SAM paper, in which they compare results on Imagenet for exact gradient penalty (accuracy 79\%), SAM (accuracy 78.9\%), SGD (76.8\%).  This is not to say that a finite-difference approximation to the HVP cannot be beneficial.  In fact, there are cases where it might well be: see [5], who highlight interactions between ReLU and point estimates of the Hessian which might favor a FD HVP for larger perturbation radii.
> 2. These questions are in the end orthogonal to the theory of our paper, which is concerned with coordinate independent regularization and derives, from general variational principles, a very general penalization scheme using a $\mathrm{Tr}(\mathbf{G^{-1} F})$ penalty with exponential smoothing of the FIM to estimate $\mathbf{G}$ which can be implemented in a number of practical ways, of which SGD-TRACER is the most naive and trivial to implement and which we have found works well in practice.  We already discuss in our manuscript the more principled approach with approximation guarantees in which, instead of the empirical Fisher, the expectation of the actual (not empirical) FIM can be estimated with the same computational cost (see Martens [3], and [2] where this is in fact the strongly recommended approach) using an MC sample per example (for classification, the Gumbel-softmax reparameterization is used to take gradients through the expectation).  Had we simply presented results for MC based estimates of the FIM we would have had a very different discussion.

---

> ### Author Response · Authors · 2024-07-08
> **comment pt 2**
>
> Responding to your other points:
>
> >it is a bit suspicious here why this is possible and the authors do not seem to face scalability issues
>
> Regarding computational costs: the forward difference implementation of our algorithm is similar to SAM, as, like SAM, it requires an additional forward and backward pass.  Even the naive implementation, double backpropagation (taking gradients through the gradient norm, an implicit HVP, naively implemented) only involves an additional cost of 2x the cost of computing the gradient (so 3x SGD, or 1.5x SAM).
>
> > So, G-Tracer becomes a pre-conditioned SGD, and the pre-condition matrix depends on the Hessian -- which makes G-Tracer closer to a 2nd-order optimizer rather than SAM
>
> Both SAM and G-Tracer use 2nd order information, explicity or implicily, for ascent, but not for descent.  Yang's construction [1], which generalizes SAM, amounts to alternating gradient descent and ascent steps, where the ascent step is an approximate second order method, (and close to exact for the small perturbation radii used in practice).  All these methods are first order methods in their descent steps, and use second order information to adjust the gradient away from a pure descent direction.
>
> >As I said in the review, I'm not saying the derivation is unacceptable; but you are using an approximation in the derivation that is not implemented in the proposed algorithm, and currently that derivation is the only justification of the algorithm -- this is what I call stretchy}
>
> The theoretical justification for all of this is the rigorous derivation of the penalty $\mathrm{Tr}(\mathbf{G^{-1} F})$ from general variational principles, using natural gradient descent in the space of variational parameters.  The implementation detail of whether to use an empirical Fisher (leading to a preconditioned gradient penalty) or an approximate Fisher or any other Hessian or Hessian Trace estimator (e.g. Hutchinson trace estimator, KFAC, etc) is not part of the derivation or formal justification of G-TRACER.
>
> Having said that, implementation details are, of course, important, and a constructive way forward suggested by our stimulating discussion is to change the presentation to put on an equal footing alternative implementations of $\mathrm{Tr}(\mathbf{G^{-1} F})$.
>
> 1) SGD-TRACER, which amounts to a preconditioned gradient penalty
> 2) using a discrete approximation to the HVP, exploring how results vary with different $\delta$
> 3) using the FIM rather than the empirical FIM (for theoretical reasons, this would be expected to produce superior results, in general)
>
> We will also add experimental results comparing the above and some discussion higlighting all the issues raised in our exchanges.  We will also provide experimental evidence on ReLU vs GELU for larger perturbation radii for these choices.  We hope that this, together with our replies here, will adress your concerns, and thank you again for a productive dialogue.
>
> [1] Penalizing Gradient Norm for Efficiently Improving Generalization in Deep Learning, Yang et al.,  Proceedings of Machine Learning Research, 2022
>
> [2] Limitations of the Empirical Fisher Approximation, Kunstner et al., NeurIPS, 2019
>
> [3] Martens, New Insights and Perspectives on the Natural Gradient Method, JMLR, 2020
>
> [4] Andrei, Accelerated conjugate gradient algorithm with finite difference Hessian/vector product approximation for unconstrained optimization, Journal of Computational and Applied Mathematics, 2009
>
> [5] How Hessian Structure Explains Mysteries in Regularization, Dauphin et al., 2024 https://openreview.net/pdf?id=Gl4AsqInti

---

### Review · Reviewer_bteo · 2024-06-17

**Summary Of Contributions:**

This paper presents a new regularizer technique, called G-TRACER, to improve the generalization performance on training deep neural nets.  This method is motivated by the Fisher information metric and the Hessian of the loss, focusing on promoting solutions with low mean curvature. The paper then demonstrates how G-TRACER can be incorporated into existing optimizers like SGD and Adam with minimal changes and without extensive tuning.

**Audience:**

Yes

**Broader Impact Concerns:**

No concern

**Claims And Evidence:**

Yes

**Requested Changes:**

The derivation in Section A.1.2, in particular, is challenging to follow, especially for those not specialized in this area.

- The paper employs two distinct uses of the Fisher Information Matrix: one within the KL trust region (equation 42) and another in approximating the Hessian (equation 54). This dual usage creates confusion about their roles and relationships.
- Following equation 46, the authors suggest that the update equation simplifies when  $p_\phi$  is parameterized as $ \phi = (\mu, \Lambda^{-1}) $. However, the subsequent argument seems to diverge from the trust-region approach discussed earlier. It would be beneficial to explicitly state the update equation for a general distribution  $q_\phi$ .
- The main update for  $\mu$  appears to involve a gradient descent step on the expected loss. It’s not immediately clear how this ties into the trust-region update described earlier on page 15. A more detailed connection between these concepts would be helpful.
- The transition from equation 52 to equation 55 is not straightforward. The substitution of  $u$  with  $w$  should be justified or explained to avoid confusion.
- On page 17, the claim that the Hessian can be approximated by the Fisher Information Matrix requires further justification. This might not be intuitive for a broader audience, and a deeper explanation would be valuable.
- More details on the derivation of the gradient in equation 71 are needed. Specifically, it is unclear how Bonnet’s theorem is applied and which aspects of (Khan & Rue) are utilized to reach the result. A step-by-step clarification would greatly improve this section’s accessibility.

Notation and Clarity:
 - On page 6, the notation $ q(w) \sim \mathcal{N}(\mu, \sigma^2 I) $ seems problematic as it includes  w  on both sides of the equation. This should be corrected or clarified to avoid confusion.

**Strengths And Weaknesses:**

Strengths

The use of geometric principles to define and implement the regularization scheme appears to be an interesting direction.

Weakness

Despite the provided intuitions, the theoretical development and motivations behind G-TRACER remain somewhat opaque.

---

> ### Author Response · Authors · 2024-07-03
> **Rebuttal**
>
> We thank the reviewer for the very helpful comments, for which we are very grateful.  We have updated the manuscript to reflect them.
>
> Taking each of the comments in turn:
>
> 1. We have clarified this in the text.  Part of the confusion stems from not having sufficiently set out the differences between the variational space FIM and the log-likelihood FIM.  We have added significantly to the "Problem setting" section to fully clarify these distinctions.
> 2. Agreed.  We have tidied this up in the new manuscript and fully reworked the derivation sketch to provide a step-by-step account.  The trust-region section is simply intended to provide background on the natural gradient methods, and doesn't play a role in subsequent derivations.  This should now be clear in the restructured text.
> 3. See 2.
> 4. Agreed.  This is a key and extremely helpful observation.  We have fully explained this step (MAP estimation), and tightened up the notation.
> 5. Agreed.  We have added a section on this, quoted the relevant Theorem and added a reference to the literature.
> 6. Agreed.  We have added details on the Bonnet and Price theorems to make this section self-contained.
>
> Regarding notation:
>
> 7. That is indeed a typo.  Thanks, corrected.

---

### Decision · Action_Editor_fX1c · 2024-07-23

**Recommendation:** Reject

**Comment:**

Despite a positive rebuttal and reviewers acknowledging improvements in the revised version, the paper does not garner a consensus for acceptance. Two reviewers lean towards rejection, while one rejects it. Concerns persist regarding the theoretical framework and clarity of contributions, as highlighted by comments such as "There are still issues with the theoretical constructions and framing of the paper" and "Confusing exposition, contributions not clear."  I strongly recommend that the authors revisit their paper, taking into careful consideration the reviewers' feedback, e.g.,  comments from reviewers "K7N4" and "Vcx7"

**Audience:**

Yes. The topic of the paper is interesting to TMLR readers.

**Claims And Evidence:**

Unfortunately, all reviewers expressed concerns regarding the "Claims And Evidence" section, indicating a lack of confidence in the validation of the paper's results.